# Arctic sea ice anomalies during the MOSAiC winter 2019/20

**Klaus Dethloff[1], Wieslaw Maslowski[2], Stefan Hendricks[3], Younjoo J. Lee[2], Helge F. Goessling[3], Thomas Krumpen[3], Christian Haas[3], Dörthe Handorf[1], Robert Ricker[3], Vladimir Bessonov[4], John J. Cassano[5], Jaclyn Clement Kinney[2], Robert Osinski[6], Markus Rex[1], Annette Rinke[1], Julia Sokolova[4], Anja Sommerfeld[1]**

[1] Alfred Wegener Institute, Helmholtz Centre for Polar and Marine Research, Telegrafenberg A45, 14473 Potsdam, Germany
[2] Department of Oceanography, Graduate School of Engineering and Applied Sciences, Naval Postgraduate School, Monterey, CA 93943, US [3] Alfred Wegener Institute, Helmholtz Centre for Polar and Marine Research, Am Handelshafen 12, 27570 Bremerhaven, Germany
[4] Arctic and Antarctic Research Institute, Center Ice and Hydrometeorological Information, Bering Street 38, St. Petersburg, Russia
[5] Cooperative Institute for Research in Environmental Sciences, National Snow and Ice Data Center and Department of Atmospheric and Oceanic Sciences, University of Colorado, Boulder, US
[6] Institute of Oceanology, Polish Academy of Sciences, Sopot 81712, Poland

Correspondence to: Klaus Dethloff (Klaus.Dethloff@awi.de) and Wieslaw Maslowski (maslowsk@nps.edu)

**Abstract.** As the Multidisciplinary drifting Observatory for the Study of Arctic Climate (MOSAiC) project went into effect during the winter of 2019/2020, the Arctic Oscillation (AO) has experienced some of the largest shifts from a highly negative index in November 2019 to an extremely positive index during January-February-March (JFM) 2020. The permanent positive AO phase for the three-month period JFM 2020 is accompanied by a prevailing positive phase of the Arctic Dipole (AD) pattern. Here we analyse the sea ice thickness (SIT) distribution based on CryoSat-2/SMOS satellite derived data augmented with results from the hindcast simulation by the fully coupled Regional Arctic System Model (RASM) for the time period from November 2019 through March 2020. A notable result of the positive AO phase during JFM 2020 were large SIT anomalies, up to 1.3 m, which emerged in the Barents sea (BS), along the northeastern Canadian coast and in parts of the central Arctic Ocean. These anomalies appear to be driven by nonlinear interactions between thermodynamic and dynamic processes. In particular, in the Barents- and Kara seas (BKS) they are a result of an enhanced ice growth connected with cold temperature anomalies and the consequence of intensified atmospheric-driven sea ice transport and deformations (i.e. ice divergence and shear) in this area. The Davies Strait, east coast of Greenland and BS regions are characterised by convergence and divergence changes, connected with thinner sea ice at the ice borders with enhanced impact of atmospheric wind forcing. Low-pressure anomalies, which developed over the Eastern Arctic during JFM 2020, increased northerly winds from the cold Arctic Ocean to the BS and accelerated the southward drift of the MOSAiC ice floe. The satellite-derived and simulated sea ice velocity anomalies, which compared well during JFM 2020, indicate a strong acceleration of the Transpolar Drift relative to the mean for the past decade, with intensified speeds up to 6 km/day. As a consequence, sea ice transport and deformations driven by atmospheric surface wind forcing accounted for bulk of SIT anomalies, especially in January 2020 and February 2020. RASM intra-annual ensemble forecast simulations with 30 ensemble members, forced with different atmospheric boundary conditions from November 1, 2019 through April 30, 2020, show a pronounced internal variability in the sea ice volume, driven by thermodynamic ice growth and ice melt processes and dynamic surface wind impacts on sea ice formation and deformation. A comparison of the respective SIT distribution and turbulent heat fluxes during the positive AO phase in JFM 2020 and the negative AO phase in JFM 2010 corroborates the conclusion, that winter sea ice conditions of the Arctic Ocean can be significantly altered by AO variability.

## 1 Introduction

The temporal evolution of Arctic sea ice thickness distribution is the result of complex and highly variable interactions within the pack ice and its interactions with atmospheric and oceanic processes (e.g. *Belter et al. 2020*). Since the late 1970s, remotely sensed measurements have provided Arctic-wide information about its changing sea ice cover (https://nsidc.org/cryosphere/seaice/study/remote_sensing.html), which motivated the development of new satellite products (*Zwally et al. 2002; Stern and Moritz, 2002; Spreen et al. 2008; Tilling et al. 2018; Neumann et al. 2019*) as well as regionally-focused coupled Arctic system models and sea ice predictions systems (e.g. *Dorn et al. 2007; Dorn et al., 2009; Maslowski et al. 2012*), to address stakeholder needs for information related to shipping, resource extraction and climate monitoring. Oceanic heat inflows into the Arctic Ocean, through the Bering Strait from the Pacific side and through the Barents Sea (BS) and Fram Strait from the Atlantic side, and sea ice impacts on the vertical structure of the upper halocline downstream. *Schlichtholz (2019)* demonstrated, that more than 80% of the variance of the leading variability mode in the winter Arctic sea ice concentration from 1981–2018, with main centers of action appearing in the BS region, can be explained by the preceding summertime temperature anomalies of Atlantic water inflow from the Norwegian Sea. The variability of Arctic sea ice distribution, drift and deformation is connected to atmospheric circulation patterns and cyclonic systems with impacts on the dynamical ice redistribution and thermo-dynamical sea ice growth- and melt. Wind patterns affect the BS and Barents-and Kara seas (BKS) ice variability through momentum transfer, advection of cold and dry or warm and humid air, forcing of warm Atlantic water inflow into the BS and by increased or decreased turbulent surface heat fluxes.

Since the BS is a shallow marginal sea, the wind-driven circulation together with tidal mixing effectively remove the bulk of Atlantic water heat to the atmosphere and only a small amount of the heat enters the deep Arctic Basin (*Gammelsrod et al. 2009*; *Onarheim et al. 2015*). Therefore, oceanic heat convergence and atmospheric winds appear as the main drivers of the BS ice cover evolution. Northerly winds influence sea ice advection mainly in winter during strong wind events and processes related to large-scale atmospheric circulation patterns, cyclonic activity, the length of the freezing season, and the remaining sea ice volume after the summer melt season are of also of importance for sea ice variability in winter.

The observed decline of Arctic sea ice was identified as a main contributor to changes in the large-scale Arctic Oscillation (AO) pattern and mid-latitude climate changes during winter, e. g. *Cohen et al. (2014)*. The origin of AO changes between positive and negative phases has been attributed to declining sea ice in Arctic regions (*Screen et al., 2013*), planetary-synoptic circulation adjustment processes (*Dethloff et al. 2006; Sokolova et al., 2017*), changes in Siberian snow cover (*Cohen et al., 2012*), weakening and warming of the stratospheric polar vortex (*Kim et al., 2014*), natural variability (*McCusker et al., 2016*) and anthropogenic greenhouse gases (*Johannessen et al., 2004*). As pointed out by *Ding et al. (2019)*, Arctic sea ice changes non-uniformly under the influence of multiple internal or external factors.

The BS has been considered as a key region for the observed fast Arctic climate changes due to intense air-sea interaction as pointed out by *Smedsrud et al. (2013)* and anomalous turbulent heat fluxes by impacting the AO winter phase via mediation of surface heat fluxes at the ocean-atmosphere interface (*Liptak and Strong 2014*). An inflow of warm Atlantic water influences the sea ice cover in the BS and its decline there has been connected to a northward shift of the Gulf Stream front *(Sato et al. (2014)*. Despite many model efforts, so far no consensus has been reached with regard to the connection of Arctic sea ice reductions with AO phase changes, with some studies pointing to positive AO changes *(e.g. Orsolini et al. 2014)* while others reasoning for negative changes *(Peings and Magnusdottir 2014)*. *Nakamura et al. (2015)* showed, that a stationary Rossby wave response to sea ice reduction in the BS might introduce anomalous circulation pattern similar to the negative AO phase and tropospheric cyclonic anomalies over Siberia, formed by the Rossby wave response to a wave source in the BKS region. *Nie et al. (2019)* emphasised the role of initial stratospheric conditions and wind anomalies in November. Westerly wind anomalies result in positive AO winter phases and the reverse happens for easterly initial anomalies. *Kolstad and Screen (2019)* showed that the correlation between autumn BKS ice and the winter North Atlantic Oscillation is non-stationary and contains considerable decadal variability. They argued that recent observed high correlation can be explained purely by internal

variability, a view supported by *Blackport et al. (2019)*. *Gong et al. (2020)* emphasized an Arctic wave train propagating from the subtropics through mid-latitudes into the Arctic and back into mid-latitudes, which is recharged and amplified in the Arctic

through anomalous surface heat flux anomalies over the Greenland Sea and the BKS. The processes responsible for the observed sea ice loss in the Arctic are influenced by coupled, nonlinear atmosphere-ocean-sea ice feedbacks in different regions of the Arctic Ocean basin as discussed by *Bushuk et al. (2019)*. The two-way interaction between ocean, sea ice and atmosphere impacts via surface turbulent heat fluxes on the lower troposphere, which feeds back with changed thermo-dynamical ice growth conditions and atmospheric wind stress forcing. *Zhao et al. (2019)* described positive and negative feedbacks related

to the AO revealed by surface heat fluxes in the Nordic Seas based on NCEP reanalysis data.

*Platov et al. (2020)* noted three modes of the surface wind forcing on the Arctic sea ice. The first, oceanic mode, is associated with the cyclonic or anticyclonic type of circulation in the Arctic Ocean as discussed by *Proshutinsky and Johnson (1997)*. The second, dipole mode, accelerates or slows down the Transpolar Drift. The third, Atlantic mode, weakens or intensifies the cyclonic gyre in the northern Northern Atlantic, corresponding to the atlantification trends (*Barton et al., 2018*)) in the BKS.

*Wang et al. (2021)* studied the impact of atmospheric wind forcing on Arctic sea ice characteristics through simulations with a coupled ocean-sea ice model and identified spatial sea ice patterns connected with AO, Arctic Dipole (AD) and Beaufort High modes.

*Trofinov et al. (2020)* described temperature decrease of more than 1°C in the inflowing Atlantic water to the BS since 2015 and argue that lower temperatures, in combination with reduced inflow during winter have caused the increases in BS winter

sea ice observed in recent years. Sea surface temperature averaged over the southern BS dropped significantly in 2019 and its annual mean value was the lowest since 2011. In the Eurasian Basin of the Arctic Ocean, *Polyakov et al. (2020)* noticed a weakening of the ocean stratification over the halocline, which isolates intermediate depth Atlantic water from the surface mixed layer. The oceanic turbulent heat fluxes increased and were greater than 10 W/m$^2$ for the winters of 2016–2018, with significant impacts on sea ice loss in this region. These oceanic changes have the potential to increase baroclinic instability in

the early Arctic winter troposphere, which impacts on synoptic scale structures in autumn and planetary waves in late winter (*Jaiser et al. 2012*), increases Arctic storm activity and play an important role for meridional heat transport into the BKS (*Long and Perrie, 2017*).

The connection between sea ice and atmospheric circulation is critical for understanding the abrupt circulation changes, which the atmosphere and sea ice experienced during the winter 2019/20. The leading atmospheric variability pattern moved from a

below-average AO negative phase in November 2019 to a highly positive and persistent AO phase during January-March 2020. The positive AO phase in the Arctic troposphere was accompanied by cold surface temperatures and enhanced near surface wind anomalies, and connected with an exceptionally strong and persistent cold stratospheric polar vortex (*Lawrence et al. 2020*). During the MOSAiC winter 2019/20, the tropospheric wave activity and wave forcing was weak and the stratospheric vortex developed an unusual configuration, which reflected planetary waves back into the troposphere and

impacted the lower atmospheric circulation. The distribution and transport of Arctic sea ice is driven by near surface wind fields, dominated in winter by the Beaufort High, which yields an anti-cyclonic sea ice drift within the Beaufourt Gyre. Its northern branch, the Transpolar Drift, moves sea ice from the Siberian coast across the deep basin toward the Fram Strait and the Nordic seas. The positive (negative) AO is characterized by low (high) sea level pressure anomalies over the Arctic that lead to cyclonic (anti-cyclonic) atmospheric circulation anomalies (*Armitage et al. 2018*), a contracted (expanded) Beaufort

Gyre circulation (*Kwok et al. 2013*) and respective shifts of the Transpolar Drift. *Proshutinsky and Johnson (1997)* discussed the alternating appearance of cyclonic and anti-cyclonic circulation regimes of the wind driven Arctic Ocean. During cyclonic regimes, low sea-level atmospheric pressure dominated over the Arctic Ocean driving sea ice and the upper ocean counter clockwise whereas during anti-cyclonic circulation regimes, high sea level pressure dominated with clockwise circulation. Circulation structures connected to the AD pattern has been discussed by *Watanabe et al. (2006)*, *Vihma et al. (2012)* and *Lei*

*et al. (2018)* for its role in sea ice export out of the Arctic.

During the winter 2019/20, the international research project MOSAiC (Multidisciplinary drifting Observatory for the Study of Arctic Climate) used the research icebreaker "Polarstern" *(Polarstern: Alfred-Wegener-Institut Helmholtz-Zentrum für Polar- und Meeresforschung. (2017))*. This vessel is operated by the German Alfred Wegener Institute, Helmholtz Centre for Polar and Marine Research and was docked to a stable sea ice floe north of the Laptev Sea in October 2019. Following the drift pattern observed by the Russian North Pole drifting stations since 1937 (*AARI 1993, Frolov et al., 2005*) the ice floe traveled from October 2019 until July 2020 with the Transpolar Drift toward the Fram Strait. *Krumpen et al. (2020)* described the origin and initial conditions of sea ice at the start of the MOSAiC experiment. Their results showed that the sea ice within 40 km of the MOSAiC Central Observatory was younger and thinner than surrounding ice and it was formed in a polynya event north of the New Siberian Islands at the beginning of December 2018. They determined, that those sea ice conditions were due to the interplay between a high ice export in the late winter preceding MOSAiC and high air temperatures during the following summer, which yielded the longest ice-free summer period of 93 days over the Siberian shelf seas since the beginning of the records. The exchange of RV "Polarstern" crew and researchers in February/March 2020, carried out for the MOSAiC project by the Russian icebreaker "RV Kapitan Dranitzyn" was significantly influenced and delayed by heavy sea ice conditions along the MOSAiC drift in the Arctic Ocean and in the BKS. Along the "RV Kapitan Dranitzyn" cruise track, in situ sea ice thickness measurements in the frame of Arctic Shipborne Sea Ice Standardization Tool (ASSIST) were carried out. Here we diagnose and focus on the regional processes in the Arctic at the ocean-sea ice interface with the atmospheric conditions, thermodynamic sea ice growth and dynamical sea ice divergence, convergence and ice shear processes during the winter 2019/20. Although an investigation of the highly nonlinear mechanisms for the AO changes is beyond the scope of this paper, the positive AO phase during January-March (JFM) 2020 is essential for the observed sea ice changes. Radiative fluxes may be changed due to regional surface albedo changes, and other factors as clouds and water vapour, in response to an external climate forcing. As pointed out by *Hall (2014)* climate signals arising from thermodynamic warming are more credible than those arising from atmospheric circulation changes.

We based our analysis on satellite derived sea ice thickness data and output of the hindcast simulation using the fully coupled Regional Arctic System Model (RASM), with AO phase nudged above 500 hPa, to examine the spectrum of nonlinear process-driven interactions between the Arctic Ocean, sea ice and the atmosphere. As a regional climate model forced along the boundaries with realistic global atmospheric reanalysis, such as the National Centers for Environmental Predictions (NCEP) Coupled Forecast System (CFS) Reanalysis (CFSR), RASM offers a unique capability to reproduce the observed natural environmental conditions in place and time. Given such capabilities, we (i) evaluate RASM skill in reproducing the sea ice thickness distribution from the CryoSat-2/SMOS satellite derived data from November 2019 until March 2020, (ii) diagnose the observed evolution of sea ice, (iii) investigate the mechanisms of and the interplay between the thermodynamic growth and dynamic sea ice processes for a positive AO phase. The synthesis of sea ice thickness distribution and growth simulated by RASM with the CryoSat-2/SMOS data, allows for improved understanding of the regional drivers of sea ice changes within the positive AO variability pattern in winter 2019/20 determined from the European Reanalysis data ERA-5. In chapter 2 we provide details of the satellite derived data and the model setup for the hindcast and forecast simulations. Chapter 3 presents results on the AO phase changes from November 2019 until March 2020 based on ERA-5 data, sea ice thickness estimates from the CryoSat-2/SMOS and the RASM hindcast, the evaluation of RASM simulations, the thermodynamic and dynamic contributions to the observed sea ice anomalies, and changes in the Transpolar Drift. We end this chapter with results from the RASM ensemble forecasts to quantify the strength of internal variability driven by regional processes within the Arctic climate system and a comparison of RASM sea ice conditions and turbulent surface heat fluxes between the AO positive 2019/2020 and AO negative 2009/2010 winters.

## 2 Data and model set up

The algorithms and methods used for the satellite retrieval of sea ice thickness products, the RASM model and the ERA5 data will be described in the following section. Monthly gridded sea-ice thickness information from remote sensing is based on the European Space Agency (ESA) CryoSat-2/SMOS Level-4 sea ice thickness data set, assessed from the Alfred Wegener Institute, Helmholtz Centre for Polar and Marine Research; https://earth.esa.int/eogateway/catalog/smos-cryosat-l4-sea-ice-thickness. The data are based on merging two independent sea-ice thickness data sets from CryoSat-2 (*Hendricks et al., 2020*) and SMOS (*Tian-Kunze et al., 2014*) by optimal interpolation (*Ricker et al., 2017*), resulting in sea-ice thickness information free from gaps in the complete northern hemisphere with sensitivity e. g. to snow cover depth across the full sea-ice thickness spectrum.

The concept is described by *Ricker (2020)* in the CryoSat2-SMOS merged product description document. An optimal interpolation scheme (OI) has been used, that allows the merging of datasets from diverse sources on a predefined analysis grid. The data are weighted differently based on known uncertainties of the individual products and an estimated correlation length scale. OI minimizes the total error of observations with respect to a background field and provides ideal weighting for the observations at each grid cell. The background field consists of a weighted average of CryoSat-2 and SMOS data two weeks before and after the rolling observation period with a length of 7 days. The CryoSat2-SMOS product is then defined as the sea-ice thickness analysis fields of the 7 day observation period with the center date as the reference time of each file. Melting does not allow to retrieve sea-ice thickness estimates from CryoSat-2 and SMOS during summer between May and September. Therefore, the merged product is limited to the period from mid-October to mid-April only due to the background field requirement.

Here, we use the product version 2.02 (*Ricker, 2019*), which is available as daily-updated gridded product with a moving observation time window of seven days between October 15 and April 15 of winter seasons since November 2010. We compute monthly sea-ice thickness fields by attributing the reference time, defined as the center time in the seven-day period, to the calendar month and average all thickness fields within one calendar month respectively. We also compute the sea-ice thickness anomaly, the difference of a monthly sea-ice thickness field, to the average conditions of each month in the CryoSat-2/SMOS data record (2010-2019) both as a difference in meters and the relative difference as a percentage of average sea-ice thickness. In addition, we use continuous, along-track, ship-based electromagnetic ice thickness measurements that were carried out on board of the Russian icebreaker "RV Kapitan Dranitzyn" during the second resupply voyage of RV Polarstern between 6 and 14 March 2020. Detailed information about the measurement principles can be found in *Haas (1998)* and *Haas et al. (1999).* Regional climate models offer exceptional spatio-temporal coverage and insights into processes and feedbacks not fully resolved in global Earth System models. They form part of a model hierarchy important for improving regional climate predictions and projections. The Regional Arctic System Model (RASM) has been developed and used to better understand the past and present operation of the Arctic climate system at process scales and to predict its change at time scales from days up to decades, see *Maslowski et al. (2012), Cassano et al. (2017),* and *Roberts et al. (2018).* RASM is a high-resolution, limited-area, fully coupled climate model, consisting of atmosphere, ocean, sea ice, marine biogeochemistry, land hydrology and river routing components. The model domain is pan-Arctic, as it covers the entire marine cryosphere of the Northern Hemisphere, terrestrial drainage to the Arctic Ocean and its major atmospheric inflow and outflow pathways, with optimal extension into the North Pacific/Atlantic to model the passage of cyclones into the Arctic. Its pan-Arctic atmosphere and land component domains are identical and configured on a 50-km grid. The ocean and sea ice components use a $1/12^\circ$ (~9.3 km, i.e. eddy-permitting) grid in both horizontal direction and 45 vertical layers. The regional model hindcast simulation was set up in the following way. The initial boundary conditions in the ocean and sea ice were derived from the stand-alone ocean and sea ice model 32-year (1948-1979) spin-up forced with the Coordinated Ocean-ice Reference Experiments phase II (CORE II, *Large and Yeager, 2008*) inter-annual atmospheric reanalysis. The ocean lateral boundary conditions were derived from the monthly University of Washington Polar Science Center Hydrographic Climatology version 3.0 (PHC3.0, *Steele et al., 2001*). The atmospheric lateral

boundary forcing as well as the grid point nudging of temperature and winds from 500 hPa to 10 hPa were based on 6-hourly NCEP CFSR data for 1979 through March 2011 and CFS version 2 (CFSv2) analyses afterwards. The hindcast simulation used here started in September 1979 and has been updated through 2020.

The RASM ensemble forecast simulations have been produced monthly since January 2019, with each ensemble (consisting of 28-31 members) initialized on the 1st of each month and run for 6 months to produce intra-annual forecasts of the Arctic environment (https://nps.edu/web/rasm/predictions). The ensemble forecasts used here were initialized on 1 November 2019 and finished by 1 May 2020. These forecasts use global output from the NCEP CFSv2 operational 9-month forecasts initialized at 0000 each day of the preceding month, resulting with the November 2019 ensemble consisting of 30 members. The RASM ensemble forecast simulations were carried out from 1st November 2019 through 30th April 2020. Each ensemble member was initialized with the same sea ice and ocean conditions and on the same date, but then it was forced by a different, 24-hr apart, NCEP forecast data set which was initialized at 00.00 between 1st and 31st October 2019. The 30-member RASM ensemble was forced with different lateral boundary conditions from 9-months forecast of the NCEP climate forecast system, applying a linear nudging of temperature, zonal- and meridional wind above 500 hPa.

For additional atmospheric analysis, ERA5 data over the Arctic region, described by *Hersbach and Dee (2016)* were used. ERA5 has several improvements compared to ERA-I as a result of higher temporal and spatial resolutions and more consistent sea surface temperature and ice concentrations.

## 3. Results

### 3.1 Analysis of atmospheric and sea-ice conditions based on ERA5 and satellite data

#### 3.1.1. Atmospheric circulation and states of the AO and AD pattern

The AO index is the leading pattern of the mean height anomalies at the surface, and a positive AO index means a lower than normal pressure in the Arctic and higher pressure outside. Figure 1a presents daily values of the AO index in mean sea level pressure (SLP) based on ERA-5 from October 2019 until May 2020 with 7-day running mean (red line) and Fig. 1b the spatial AO pattern north of 20 °N. The AO pattern was defined as the leading mode of Empirical Orthogonal Function analysis of monthly mean SLP during the 1979-2000 period over the domain 20°-90°N. This domain and reference period was used for the calculation of the spatial AO patterns to ensure comparability with the widely used AO index provided by the NOAA Climate Prediction center (CPC, https://www.cpc.ncep.noaa.gov/products/precip/CWlink/daily_ao_index/ao.shtml), which is based on the AO pattern calculated for the mentioned reference period and NCEP/NCAR reanalysis data set. The daily AO indices (Fig. 1a) have been obtained by projecting ERA5 daily SLP data from 1979 to May 2020 onto the AO pattern shown in Fig. 1b. For comparison the loading pattern of the AO for the different ERA-5 reference period 2010-2019 was computed (not shown), and the corresponding AO index for the MOSAiC period, obtained by projecting the daily SLP anomalies onto this loading pattern. The time series of daily values of the AO index from October 2019 to April 2020 obtained by projecting the daily SLP anomalies onto the loading pattern from 2010-2019 agree entirely with Fig 1a.

The shift from a negative phase in November to the positive AO phase in January, February and March 2020 is displayed in Figure 1. Figure S1 presents the PDFs of the daily AO indices for November and January-March (JFM) from 1979 until 2018/2019 (gray) in comparison to November 2019 (blue) and JFM 2020 (blue) with the prevailing positive AO-index in 2020. Figure 2 displays the SLP anomaly and the 2 m temperature anomaly for November 2019 and January 2020 and the SLP anomalies for February 2020 and March 2020 compared to the mean for 2010-2019 based on ERA-5 data. During November the negative AO phase occurs with higher pressure anomaly over most regions of the Arctic Ocean and relatively warm temperatures in the Beaufort and Siberian seas. This circulation is connected with atmospheric 10 m winds from the south-west of Greenland and warm air masses inflow into the Western Arctic. During January 2020 a low-pressure anomaly developed over the Eastern Arctic and a high-pressure anomaly over the Western Arctic. This atmospheric flow configuration induced strong northerly winds from the cold Arctic Ocean to the BS and accelerated the southward drift of the MOSAiC ice

floe in the Transpolar Drift. A regional cold temperature anomaly developed in the northern part of BS. In February 2020 the low-pressure system stayed over the BKS and adjacent land regions and pushed sea ice into the BS, whereas the Kara Sea experienced southerly winds and thus warm anomalies. By March 2020 the low-pressure anomaly was located north of the Laptev Sea, inducing westerly wind anomalies following Arctic cyclone tracks in the BKS and keeping the cold air in the Arctic.

Besides the AO, the Arctic Dipole (AD) pattern is important for the Arctic circulation and sea-ice motion (*Wu et al., 2006; Cai et al., 2018; Watanabe et al., 2006; Zhang, 2015*). The AD pattern in its positive phase is connected with a negative pressure anomaly over the eastern Arctic and a positive pressure anomaly over the western Arctic and leads to an acceleration of the transpolar drift in agreement with *Lei et al. (2016)*. Previous studies on the AD pattern, either in summer (*Cai et al., 2018*), or in winter (*Wu et al., 2006*) often used a rather small domain (60°-90°N or 70°-90°N) and defined the AD pattern as second EOF of monthly mean SLP fields. For these small areas the domain boundaries do induce an artificial preference of particular pattern structures as discussed by *Legates (2003)* and *Overland and Wang (2010)*. Since neither EOF2 nor EOF3 of the above described analysis for the large domain 20°-90°N reveal an AD pattern, an additional EOF analysis of monthly mean SLP over the smaller domain 60°-90°N was performed, over the same 1979-2000 period as before.

Fig. S2 shows the respective first three EOFs and their daily indices. The first EOF displays again the AO pattern, and the daily indices over the MOSAiC period Nov 2019 to May 2020 are highly correlated (0.95) with the AO index based on the EOF1 for the large domain (Fig. 1a) In this analysis, the AD pattern appears as third EOF, which indicates that the AD pattern is less stable than the AO pattern. The explained variances are 15 % for the second EOF and 13.6 % for the third EOF (AD). The positive AO phase from January-March 2020 is accompanied by a prevailing positive phase of the AD pattern (Fig. S2). The histogram of the daily AD indices for the period January to March 2020 indicates a higher variability of the AD index compared to the AO index (compare Fig. S1, right and S3). Whereas the AO index remain positive over the whole period January to March 2020, the AD index shows a prevailing positive phase, but with a smaller shift of the distribution towards positive values compared to the shift in the distribution of the AO index. The time series of the AD index reveals more positive values in January and March, but a shift to more neutral and negative values in February (see time series for EOF3 in S2). This behavior of the AO and AD indices explains to a large extent the differences in the monthly mean SLP pattern over the Arctic for January, February and March, displayed in Fig. 2.

*Krumpen et al. (2021)* analysed ship-borne observations of winds, air temperatures and sea level pressure along the MOSAiC ice drift trajectory with ERA-5 data for the time period 2005-2020. Fig. S4 compares the 10 m wind, 2m temperature and sea level pressure along the MOSAiC drift trajectory based ERA-5 data for the climatology 2010-2019 applied in this study. Strongest deviations from the climatology occur in the time period January-March 2020. In mid-February a low surface pressure anomaly is determined by a strong synoptical cyclone event with values down to 985 hPa. This low pressure anomaly is connected with warmer temperatures and higher wind speed. Contrary high pressure values at the beginning of March 2020 are connected with cold temperatures and lower wind speed, indicating the important role of warm or cold advection for temperature changes.

*Lei et al. (2016)* investigated the sea ice motion from the central Arctic to the Fram Strait with ice-tethered buoys between 1979 and 2011 and showed, that sea ice drift was determined mainly by near surface winds. They detected an accelerated meridional sea ice velocity following the Arctic Dipole (AD) pattern and a reduced meandering of the ice trajectories during the positive AD phase. The drift of the central MOSAiC Observatory was closely correlated with the ERA5 zonal and meridional components of the 10m winds (blue and red curves in Fig. S5). Compared to previous years, winds tended to have anomalies toward the Fram Strait, in particular in January, February, and March 2020 (compare red and black curves in Fig. S5, bottom), in line with corresponding sea-level pressure patterns (Fig. 2). Moreover, while the ice drift speed amounted to about 2% of the 10m wind speed on average, the drift component toward Fram Strait was positively offset compared to the winds. In particular from mid-February until the end of March, several short periods of wind toward eastern Siberia (negative

values in Fig. S5, bottom) did not result in accordingly reversed drift, but only prompted the transpolar drift to pause (values close to zero in Fig. S5, bottom), likely due to the continued action of ocean currents and/or internal ice stress. From mid-June onwards, the ice drift was superimposed by pronounced inertial motions (Fig. S5), hinting at a looser ice cover (e.g., *Gimbert et al, 2012*). A close relation between the 10 m wind speed components and the sea ice velocity during the positive AO months January-March 2020 is visible. The MOSAiC drift showed a fast accelerated drift from the central Arctic to the Fram Strait without meandering. These results underlines, that the direct fast southward MOSAiC drift towards the Fram Strait during January-March 2020 was a result of the permanent positive AO phase accompanied by a prevailing positive AD phase.

### 3.1.2 Sea ice thickness and extent

Sea ice thickness and their anomalies for November 2019 through March 2020 based on CryoSat-2/SMOS satellite data analysis compared to the mean condition in the entire data record (2010-2019) are presented in Fig. 3. They show a regionally varying pattern of positive and negative sea ice anomalies. In November 2019 positive thickness anomalies occur in the Beaufort Sea, BS and northeast of Spitsbergen. At the Bering Strait a negative ice anomaly exists. In December 2019 (not shown) already weak sea ice anomalies to start develop in the BKS. In January 2020 a pronounced ice anomaly is visible in the BKS, which persisted with regional changes in the Kara Sea through February into March 2020, when sea ice thickness increased west of Spitsbergen. Positive ice anomalies developed at the Bering Strait and the Canadian coast. In relative terms, the anomaly in the BKS is more significant, as it almost doubled the thickness in the first-year ice region as seen in relative sea-ice thickness anomaly fields in the third column of Figure 3. November 2019 was a month with pronounced negative AO phase, whereas the months JFM 2020 were marked by a strong positive AO. Figure 3 shows enhanced sea ice anomalies in the BKS during JFM 2020. These sea ice anomalies occur at the same time as the persistent positive AO phase (Fig. 1). To understand the underlying thermo-dynamic and dynamic contributions for the observed sea ice thickness evolution we will discuss simulation results from the fully coupled RASM model.

### 3.2 Simulation of atmospheric and sea ice conditions in RASM
### 3.2.1 Model evaluation

Figure S6 presents the RASM-simulated atmospheric large-scale circulation exemplarily for January 2020 which compares well to the SLP anomalies in the positive AO phase shown in Figure 1. The pronounced negative 2m temperature anomaly observed in the BS, seen in Figure 2 for January 2020, is also reproduced. The accurate simulation of this atmospheric circulation pattern is a result of grid point nudging of the atmosphere above 500 hPa in RASM to the AO phase. The SLP and temperature anomalies simulated by RASM (Figure S6) are associated with positive SIT anomalies in the BS and east of Spitsbergen presented in the second and third row of Figure 4. The simulated ice thickness anomalies for November 2019, January, February and March 2020 are in qualitative agreement with the satellite derived SIT anomalies for the same months displayed in Figure 3. The largest positive thickness anomalies between 1.0 and 1.5 m occur in the BS, along the north-eastern Canadian coast and in the central Arctic Ocean. In all other regions and especially over the Bering Strait and the Siberian Seas the sea ice is thinner than the 2010-2019 mean. The RASM simulations (Fig. 4) also exhibit positive SIT anomalies of the Arctic Ocean north-west of Greenland, not found in the CryoSat-2/SMOS-derived SIT anomalies (Fig. 3). A more in-depth comparison of the SIT for November 2019 and JFM 2020 (Fig. 5) indicates thicker ice stretching further into the central basin from Greenland in the CryoSat2/SMOS-derived SIT compared to the RASM. In the BKS, the Laptev Sea and the Bering Strait the RASM simulations indicates thicker sea ice in the range of up to 1 m compared to the satellite-derived data. The SIT simulations in RASM has been independently compared in a quality control with other coupled and uncoupled model systems by *Roberts et al. (2018)* and are in good agreement with the limited observations. A high correlation between CryoSat2/SMOS-derived SIT and RASM simulations is visible in the right column of Fig. 5. The differences in SIT (Fig. 5) may be partly

connected to the impact of surface roughness on the radar freeboards and the retrieval algorithms as discussed by *Landy et al. (2020)*.

The merged CryoSat-2/SMOS SIT data is dominated by the CryoSat-2 radar altimeter contribution in areas with multi-year sea ice. Radar freeboard is the term in sea ice radar altimetry that describes the height of the ice surface above local sea level perceived by a radar altimeter. It differs from sea ice freeboard, the actual height of the ice surface, by a correction that requires prior knowledge of snow depth and density. The purpose of this correction is to remove the impact of the slower wave propagation speed of the radar pulse within the snow lawyer on the radar range and thus ice surface elevation. The corrected radar freeboard is then converted to sea ice thickness using information of the densities of sea ice, ocean water, and snow, and estimating the depth of snow accumulated on the ice surface based on climatological values (Hendricks et al. 2020). In *Landy et al. (2020)* it is demonstrated that sea ice surface roughness may cause a systemic radar freeboard uncertainty which represents one of the principal sources of pan-Arctic SIT uncertainty. In the CryoSat-2 retrieval algorithm of the CryoSat-2/SMOS SIT data set this systemic bias might contribute to the higher CryoSat-2/SMOS thicknesses in the central Arctic and specifically north of the Canadian Archipelago with respect to RASM in Fig 5. But this assertion does not consider other systemic uncertainties present in the CryoSat-2 retrieval such as the underestimation of sea ice density for multi-year ice in recent years (*Jutila et al. 2021*), which might compensate the radar freeboard bias to an unknown extent. In the comparison to other SIT data sets, CryoSat-2/SMOS also yields thicker ice in the central Arctic compared to ICESat-2 estimates and though these difference are within the range of the SIT uncertainty resulting from different retrievals, an indication of SIT overestimation by CryoSat-2/SMOS remains.

For the construction of Fig. 5 the 50% threshold method was used, which indicates thicker ice in the central Arctic compared to the ICESat-2 estimates. *Landy et al. (2020)* showed, that variable ice surface roughness contributes a systematic uncertainty in sea ice thickness of up to 20% over first-year ice and 30% over multiyear ice, and represents one of the principal sources of pan-Arctic sea ice thickness uncertainty.

RASM skill is assessed based on root-mean-square difference (RMSD) against observational data. Figure 6a shows the Target diagram (*Joliff et al., 2009*) to display unbiased RMSD (uRMSD; x-axis) and bias (y-axis) for monthly SIT on a single plot: i.e., $RMSD^2=bias^2+uRMSD^2$. These quantities are normalised by the standard deviation of CryoSat2/SMOS SIT. Figure 6b is the Taylor diagram (*Taylor, 2001*) representing the relative skill of RASM with respect to CryoSat2/SMOS. It provides an additional set of statistics in uRMSD by displaying the correlation and the ratio of the standard deviation between RASM and CryoSat2/SMOS SIT for each month from November 2019 until March 2020.

A high correlation between satellite and model estimated SIT exists in all considered months (Fig. 5, right column). RASM bias and root mean square differences (Fig. 6a) and the standard deviations and correlations (Fig. 6b) relative to CryoSat2/SMOS for each month from November 2019 until March 2020 with respect to sea ice thickness show for all months a high correlation above 0.7 and a low standard deviation of the model simulations compared to the satellite based SIT. The monthly mean sea ice thickness, standard deviations, correlations, bias and root mean square difference between CryoSat-2/SMOS data and the RASM hindcast simulations from November 2019 until March 2020 (Tab. 1) indicate an agreement with high correlations between 0.84 and 0.86 and a low domain-averaged bias. The domain averaged bias is the difference between RASM and satellite data in a region defined in Fig. 12 including all sea-ice regions with boundaries at DS (Davis Street), FS (Fram Strait), BSO (Barents Sea Opening) and BS (Bering Strait).

### 3.2.2 Interpretation of positive sea ice anomaly in the BS

The integrated sea ice growth anomalies of RASM (Fig. 7) compared to the mean 2010-2019 indicate regionally varying ice growth over the whole Arctic Ocean during polar night conditions and regions of enhanced ice growth, which in November 2019 starts north-west of Greenland and along the sea ice border in the eastern Arctic. In January 2020 strong ice growth takes place in the BKS and the Siberian Sea and the Bering Strait. In February 2020 the strongest ice growth anomalies are simulated

over the Beaufort and Chukchi Sea and in March 2020 over the mid-Arctic Ocean and parts of the BS and at the east coast of Greenland. In February and March 2020 weaker sea ice growth is modelled in the Laptev Sea, east of Greenland and the Davis Strait.

The EM ice thickness measurements on board of the Russian icebreaker "RV Kapitan Dranitzyn" undertaken during the MOSAiC resupply voyage between 6 and 14 March 2020 indicated heavy sea ice conditions between 84° and 88°N in the BS. Mean daily, modal thicknesses are compared to the mean RASM simulations between 6 and 14 March 2020 (Fig. 8). The ship-based measurements ranged between 1.3 and 1.5 m and are 0.3-0.5 m thinner than those from RASM. The ship based sea ice thickness is also 0.3-0.4 m thinner than what was observed by ground-based measurements at the MOSAiC ice floe (not shown), where modal thicknesses between 1.7 and 1.8 m were measured. The main bias of the EM ship-based measurements is connected to difficult calibration on the ramming icebreaker. The frequent ramming operations of the ship with little progress over the undisturbed heavy ice makes processing and filtering of the ship-based measurements challenging. However, RASM results would agree very well with observations, if ~0.4 m were added to the ship-based data. Consequently, also the regional gradients in both data sets with thinner ice to the south are well described. The movement of "RV Kapitan Dranitzyn" to "RV Polarstern" in the sectors of the BS of the Arctic basin in February 2020 and March 2020 was carried out under severe ice conditions with thick first-year and second-year ice. The movement of the supply vessel slowed down significantly due to the absence of extensive leads in the meridional direction and compression (especially in February 2020 on the way to "RV Polarstern"). Large sea ice leads showed predominantly an orientation in zonal direction as a result of the positive AO. Compression weakening and local fracture system allowed "RV Kapitan Dranitzyn" to move forward gradually to "RV Polarstern". Sea ice leads in the infrared channel of NOAA-20 satellite pictures on 5 March 2020 with a resolution of 375 m points to a more zonal orientation due to the low-pressure systems connected to the positive AO phase (Fig. 9). A snapshot of the sea ice divergence on 5 March 2020 from the RASM simulations shows a qualitatively-similar orientation of sea ice leads in the BS and north of it

Comparison of Figures 4 and 7 points out, that positive sea ice growth anomalies in the BS occur in the region of positive thickness anomalies during JFM 2020. Therefore, enhanced sea ice thickness in the BS is partly a result of enhanced ice growth due to colder temperature anomalies in this area (Fig. 2 and Fig. S6). But ice thickness anomalies are influenced by deformation parameters, e.g., divergence/convergence and shear, commonly generated in response to strong and/or persistent winds. *Onarheim et al. (2015)* demonstrated, that changes of surface wind stress may explain 78% of the sea ice extent variance in the BS. This confirms that oceanic heat transport and surface wind stress are the main drivers of sea ice distribution and their variabilities capture most of the sea ice variance in the BS. The exchange of momentum due to turbulent atmospheric processes controls the sea ice motion. Divergence generates open water areas where new sea ice growth may occur. Convergence leads to the formation of pressure ridges and the SIT distribution in a region can be determined by the number and thickness of ice ridges.

The RASM-simulated positive and negative ice divergence anomalies and the ice shear anomaly for the JFM 2020 mean compared to the JFM 2010-2019 mean, together with the transpolar drift in km/day are indicated by black arrows and presented in Fig. 10. In all three plots the transpolar drift in km/day is indicated by thick black arrows. Longer black arrows in the Davis Strait, the east coast of Greenland and the BS indicate individual grid cells with a very different drift. Blue colours in the top part of Fig. 10 indicate regions with reduced convergence and red colours in the middle part of Fig. 10 indicate those with enhanced divergence. These grid cells are likely reflecting the free-drift of thinner sea ice in marginal ice zones, where the impact of atmospheric wind forcing on the ice drift is much less limited compared to the drift within pack ice.

The sea ice drift is a result of the near surface wind fields, which determines sea ice deformation as described e. g. by Spreen et al. (2011). Sea ice momentum changes are a result of air-ice and ice-ocean stresses as discussed e. g. by Martin et al. (2016). Negative ice divergence anomalies occur in the Fram Strait and the BS between Spitsbergen and Novaya Zemlya and are displayed in Fig. 10 (top). Positive ice divergence anomalies are present in the Fram Strait and the west coast of Spitsbergen

and presented in Fig. 10 (middle). This region also shows strong positive and negative values for the ice shear anomaly in Fig. 10 (bottom), which indicates the strong dynamical impact in the sea ice formation and deformation in the BKS region. Reduced divergence, which corresponds to enhanced convergence appears in a belt between west Greenland and the Kara Sea. In areas close to the ice edge, positive ice shear coincide with positive sea ice concentration anomalies, since more ice than usual exists there. The ice divergence anomalies are weaker and occur in the region of strongest sea ice growth, whereas ice shear processes due to wind stresses in Fig. 10 (bottom) indicate here positive anomaly values. The sea ice thickness anomalies in the BS region, where negative temperature anomalies occurred, are a result of wind stresses, which enabled ice growth or ice deformation. Atmospheric surface wind stresses impact sea ice deformation and the 10 m wind vectors show a strong wind components from the North in the region east of Spitsbergen (see Fig. 2). The positive AO phase and related near surface winds during JFM 2020 may be connected to an intensified and northward shifted Atlantic storm track as earlier discussed by *Serreze et al. (1997), Nie et al. (2008)* and *Inoue et al. (2012)*.

### 3.2.3. Transpolar sea ice drift

The model-simulated sea ice velocity anomaly (Fig. 11, top) and satellite-derived sea ice velocity anomaly (Fig. 11, bottom) both in km/day compared to the climate mean 2010-2019, computed from OSI-SAF low resolution sea ice motion data during January-March 2020 indicates a strong acceleration of the Transpolar Drift during the MOSAiC winter, with intensified speeds up to 6 km/day. The Ocean Sea Ice Satellite Application Facilities (OSA-SAF) deliver satellite derived scatterometer winds, sea surface temperatures and sea ice surface temperatures, radiative fluxes, sea ice concentration, edges, types and sea ice drift. The black arrows over the redish shading in the Eastern Arctic indicate the transpolar drift. Longer black arrows in the Davis Strait, the east coast of Greenland and the BS indicate the free-drift of grid cells within marginal ice zones.

This drift is in general agreement with the 10 m winds displayed for January, February and March 2020 and the low-pressure anomalies presented in Figure 2. The centers of the persistent low-pressure systems over the Arctic Ocean corresponding to the positive AO phase, changed their positions during JFM 2020. In March 2020 the center moved toward Siberia, impacted the ice drift velocities in the BS region and contributed to the increased sea ice thickness in the region around Spitsbergen. The low-pressure anomaly in March 2020 induced a stronger drift towards the BS. Sea ice growth in the BS was the combined effect of thermodynamic growth due to the colder temperatures there and dynamical SIT changes related to the positive AO phase and altered wind stresses which affect the ice divergence. The simulated sea ice velocity anomalies agree well with the satellite-derived sea ice velocity anomalies especially over the eastern part of the Arctic Ocean (Fig. 11). Over the Beaufort Sea and the western part of the Arctic Ocean the sea ice drift in the RASM simulations is underestimated and the direction differs compared to the satellite derived data.

### 3.2.4 Internal variability

Previous work by *Ding et al. (2019)* and *Nie et al. (2019)* emphasized the importance of internal climate variations for the shift of the AO phases. Here, we examine related regional sea ice variations in the Pan-Arctic and in the BS domains (Fig. 12) with the nudged AO in ensemble forecasts. The Pan-Arctic domain covers the whole Arctic Ocean with borders at the Bering Strait (BSr), the Fram Strait (FS), the Barents Sea Opening (BSO) and the Davies Strait (DS). The temporal evolution of the mean absolute difference (relative to the ensemble mean) in the simulated Pan-Arctic and BS sea ice volume for the RASM 30-member ensemble 6-month forecast simulations from November 1, 2019 through April 30, 2020 is shown in Fig. 13. Each ensemble member was initialized with the same sea ice and ocean conditions and on the same date, but then it was forced by a different, 24-hr apart, NCEP Climate Forecast System (CFSv2) global forecast initialized 24-hr apart at 00.00 between 1st and 31st October 2019. The 30-member RASM ensemble was forced with different lateral boundary conditions from 9-months forecast of the NCEP climate forecast system, applying a linear nudging of temperature, zonal- and meridional wind above 500 hPa. The differences among the 30 ensemble members for the Pan-Arctic domain are in the range of 1000 km$^3$ and show

significant positive or negative departures from the ensemble mean volume. The results presented in Figure 13 points to the large internally generated variability of the Pan-Arctic and BS sea ice volume changes in the coupled regional system and remote impacts from the mid-latitudes. Differences in modelled sea ice volume vary significantly between the Pan-Arctic and the BS region and can be even of opposite sign, e.g. as visible in ensemble member 2 and 8. The sea ice evolution distinguishes among all 30 ensemble members. e.g. ensemble members 1, 2, 3, 10, 13, 19, 25, and 26 indicate positive sea ice volume differences of different strength during the winter 2019/20 in the Pan-Arctic domain, whereas e. g. ensemble members 7, 8, 9, 15, 16, 22, and 23 show negative ice volume differences of varying strength. In the BS ensemble member 2, 4, 8, 12, 19, and 30 indicate different ice volume trends in comparison to the Pan-Arctic domain.

To quantify the underlying mechanisms for the Arctic ice volume differences and diverging temporal evolution we display the thermodynamic sea ice volume tendencies (TVT) with combined ice growth and ice melt terms for all 30 ensemble members for the Pan-Arctic and the BS domain in Figure 14 from November 2019-February 2020 and March 2020. Compared to the hindcast values (yellow bars) the tendencies in the different months varies both for the Pan-Arctic and BS domain. In February 2020 and March 2020 the mean sea ice volume tendencies reaches 73 to 58 $km^3$/day in the Pan-Arctic. The standard deviation remains similar strong from November 2019-February 2020 and becomes weaker in March 2020. The thermodynamic sea ice volume tendencies in the BS during January 2020 and February 2020 are in the range of 6 $km^3$/day and above 4 $km^3$/day during March 2020.

Standard deviations in the BS are highest in December 2019. In addition, we show the dynamical ice volume tendencies (DVT) in the BS (note that the Pan-Arctic dynamical ice volume tendencies are zero by definition), which are weaker and indicate a sea ice decline in most ensemble members and all months. Only 6 ensemble members show positive dynamical ice volume tendencies during January and February, but 8 members in March. The hindcast simulation indicates BS dynamical tendencies near-zero in January, negative in February but positive in March 2020.

The statistical properties of thermodynamic sea ice volume tendencies are based on differences of daily values and shown for the Pan-Arctic and the BS domains and dynamical ice volume tendencies for the BS region from November 2019-March 2020 (Fig. S7-S9). The strong deviations due to internally generated variability in Arctic sea ice growth and dynamical ice deformations are visible. Ensemble members 2 / 8 has been selected as the representation of maximum / minimum ice volume difference for the Pan-Arctic domain. However, those two ensemble members are not representative of the BS, which is why respective ensemble members 4 and 9 are selected. In the RASM hindcast, the Pan-Arctic thermodynamic sea ice volume tendencies increases due to ice growth from November 2019 until January 2020. The differences between the hindcast and the four selected forecast simulations 2, 4, 8, and 9 are large through all months from November 2019 until February 2020 and can reach ~20 $km^3$/day, or ~600 $km^3$/month. The thermodynamic ice volume tendencies in BS for the same four ensemble members 2, 4, 8, and 9 is in the range of 3 $km^3$/day or ~90 $km^3$/month.

The SLP anomalies in November 2019 and January 2020 (Fig. S10) for the hindcast simulation and the Pan-Arctic ensemble member 2 with positive sea ice anomaly in Fig. 13 and Pan-Arctic ensemble member 8 with negative sea ice anomaly in Fig. 13 has been selected. The SLP pattern (Fig. S10) for ensemble member 8 shows a strong low pressure anomaly in January over Siberia and that for ensemble member 2 a high pressure anomaly in January over the Arctic Ocean. The 500 hPa geopotential heights for the RASM hindcast and the two ensemble members (not shown) indicate a pronounced barotropic structure in the troposphere and emphasize the diverging development of the pressure, temperature and geopotential patterns and the important role of internally generated variability in sea ice formations as pointed out by *Ding et al. (2019).* Compared to the short daily atmospheric time scales the longer time scales of ocean and sea ice processes provide memory effects for seasonal sea ice forecasts, but the large atmospheric variability connects sea ice predictability to atmospheric wind predictions of up to 10 days as discussed by *Inoue (2020)* and set inherent limits for seasonal sea ice predictions as pointed out by *Serreze and Stroeve (2015).*

The differences of thermodynamic and dynamic ice volume tendencies per model grid cell (Fig. 15) represents the sea ice redistribution between RASM Pan-Arctic ensemble member 2 (positive ice difference in Figure 13) and Pan-Arctic member 8 (negative ice difference in Figure 13). The largest sea ice volume increase occurs in both ensemble members in the BKS and at the north-west side of Greenland (Fig. 15). In both ensemble members the thermodynamic ice growth is in the range between 0.5 and 1 m/winter with more enhanced ice growth in the Laptev Sea (not shown). The differences between the two forecast ensemble members are in the range of up to -0.5 m in the BKS and up to 0.3 m over the Arctic Ocean. The accumulated winter (JFM 2020) ice volume tendencies following dynamical and thermo-dynamical drivers are largest in the BS. Bigger ice volume differences occur north-west of Greenland with more than 0.5 m during the winter. The pan-Arctic sea ice volume represents the coupled system response to large-scale forcing and it is a better diagnostic of different sea ice regimes among ensemble members since the sea ice extent and sea ice area is relatively similar in winter. Compared with the dynamical contributions, the thermodynamic growth processes (Fig. 15) lead to greatest differences between the two ensemble members in the BKS and at the ice edge region around Spitsbergen and Greenland and exhibit similar peak differences. Although the greatest differences occur at the ice edge regions, remarkable changes in the inner Arctic north-west of Greenland are visible and sea ice volume in this region is to a large extent determined by dynamical processes.

### 3.2.5 Case study of positive and negative AO winters

To contrast the sea ice conditions and the regional processes and feedbacks for positive and negative winters AO we compare the RASM hindcast results during the MOSAiC winter 2020, with an exceptionallly positive AO phase, against the exceptionally negative AO winter of 2009/2010. Figure S11 displays the AO time series of the AO index from October 2009 until May 2010, which indicates a weakly positive AO phase in November 2009 and the strongest negative AO phase in winter 2010 during the last 60 years (*L'Heureux et al. 2010*). As discussed, e. g. by *Zhao et al. (2019)* the AO phase is closely related to sea ice variability over the Arctic Ocean. Surface heat fluxes in the coupled Arctic climate system in winter are influenced by different positive and negative feedbacks as e. g. vertical ocean convection, atmospheric turbulence, latent heat and cloud formations, long wave radiation, oceanic currents, Arctic storms and atmospheric circulations and can be considered as an integrated quantity related to all these processes. This regional approach has obvious limits, since e. g. *Gong et al. (2020)* showed the existence of a hemispheric planetary wave train propagating from the subtropics through mid-latitudes into the Arctic and back, thereby recharged and amplified over the Arctic through anomalous latent heating over the Greenland Sea and BKS.

Figure S12 shows the RASM simulations of SLP and 2m temperature for January 2010, which represents the negative AO phase, given the atmospheric nudging in RASM above 500 hPa. The nudging of the regional coupled Arctic climate system model RASM to the different AO phases allows an efficient albeit coarse diagnosis of differences between the surface heat fluxes for the two AO phases. Comparison of Fig. S12 with Fig. S6 indicates an inverse temperature anomaly pattern between the western and the eastern Arctic for positive and negative AO winters. Under positive AO conditions in winter (Fig. S6 exemplarily for January 2020)) negative temperature anomalies occur over the eastern Arctic and positive anomalies occur over the Canadian Basin in the western Arctic. During the negative AO winter conditions in January 2010 (Fig. S12) the eastern Arctic reveals weak positive temperature anomalies and the western Arctic negative temperature anomalies.

The SIT differences and thermodynamic ice volume differences between the JFM mean 2010 and JFM mean 2020 together with the turbulent surface heat fluxes for JFM 2010 and 2020 from the RASM hindcast simulations has been computed (Fig. 16), During the negative AO winter 2010 SIT was enhanced in the Beaufort- and the Siberian seas and in a belt from the north coast of Greenland to the Canadian Arctic with SIT differences greater than 1 m. In the western part of Arctic Ocean and the BKS ice thickness was weaker in JFM 2010 compared to JFM 2020. The thermodynamic ice volume tendencies indicate stronger sea ice growth over most parts of the Arctic Ocean in JFM 2010 except north of Greenland and in parts of the BKS.

555 During winter under polar night conditions the main component of the surface heat budget is through sensible and latent heat fluxes and a stronger heat release from the ocean to the atmosphere connected to the North Atlantic oceans current occur. The surface heat fluxes for JFM 2010 and JFM 2020 indicates stronger heat fluxes for the positive AO winter phase 2020 in the North Atlantic Ocean south and east of Greenland, in the western BS region and at the North-American coast on the Pacific side (Fig. 16). Negative values mean that the ocean is losing heat to the atmosphere. The difference plot in the lower row of 560 Figure 16 identifies enhanced heat flux changes in the North Atlantic around and south of Greenland, along the eastern coast of Greenland and in the western BS with values of 150 W/m$^2$ during the positive AO phase. Along the Norwegian coast and the eastern BS the turbulent heat fluxes are reduced. On the Pacific side a similar dipole pattern with enhanced fluxes along the North-American coast and reduced fluxes in the Okhotsk Sea is visible. This difference structure on the North Atlantic side agrees very well with the NCEP based surface heat flux analysis of *Zhao et al. (2019)*, who showed the high correlation 565 between the sensible heat fluxes in this North Atlantic region with the AO index (their Figure 8a). Positive surface heat fluxes in this region are positively correlated with a higher AO phase. The main factor mediating the turbulent surface heat fluxes is the meridional wind component in the Nordic Seas. *Zhao et al. (2019)* showed, that during a positive AO index the atmospheric circulation enhances the transport of warm and humid air into the Arctic along the Norwegian coast through southerly winds and the transport of cold and dry air to the Atlantic along the Greenland Sea coast via northerly winds (see their Figure 10). 570 The view of *Gong et al. (2020)* (their Figure 7f) about heating anomalies in the Greenland, BS and Kara Sea as a possible source of planetary wave activity over the Arctic Ocean may be supported by our results (Fig. 16).

## 4 Summary and Conclusions

Monthly averaged sea-ice thickness from remotely sensed data based on the ESA CryoSat-2/SMOS data set has allowed the determination of Arctic-wide sea ice thickness distributions between the 15 October 2019 and 15 April 2020. We analyzed 575 and compared the satellite derived sea ice thickness product with results from the hindcast simulation using the fully coupled RASM for the time November 2019 until March 2020. The synthesis of sea ice thickness and ice growth simulated by RASM with the CryoSat-2/SMOS data contributes to a better understanding of the regional coupled atmosphere-ocean-sea ice processes during the period with a positive AO determined from ERA5. The histograms and time series of the daily AO index and AD index indicates, that from January-March 2020 the permanent positive AO phase is accompanied by a prevailing 580 positive phase of the AD pattern.

A comparison of the SIT for November 2019 and JFM 2020 indicated thicker ice in the central Arctic in the CryoSat2/SMOS data compared to the RASM. In the BS, the Laptev Sea and the Bering Strait RASM simulates thicker sea ice in the range of up to 1 m compared to the satellite derived data. This may be partly connected to the impact of surface roughness on the radar freeboards and the retrieval algorithms discussed by *Landy et al. (2020)*. The agreement between CryoSat-2/SMOS data and 585 the RASM hindcast simulations is acceptable with high correlations between 0.74 and 0.76.

Connected to anomalous atmospheric circulation in winter 2019/20, with a positive AO phase from January-March 2020, thickness anomalies between 1.0 and 1.5 m occurred in the BS, along the north-eastern Canadian Coast and in the central Arctic Ocean. Sea ice thickness measurements with an EM instrument on board of the resupply icebreaker „RV Kapitan Dranitzyn" and thickness measured at the MOSAiC floe were in reasonable agreement with the RASM sea ice thickness 590 simulations considering an obvious bias of 0.4 m between the ship- and ground-based measurements at the MOSAiC floe. In January 2020 ice grew in the BKS and the Siberian Seas and the Bering Strait. In February 2020 strong ice growth anomalies occurred in the Beaufort Sea and in March 2020 over the mid-Arctic Ocean and parts of the BS and at the east coast of Greenland. In February 2020 and March 2020 less sea ice growth appeared in the Laptev Sea, east of Greenland and the Davis Strait. The positive sea ice thickness anomaly in the BS during January 2020 and March 2020 is a result of enhanced ice growth 595 related to the negative temperature anomalies in this area, and a consequence of intensified sea ice convergence and ice shear.

Compared with the dynamic contributions the thermodynamic growth processes lead to greatest differences in the BKS and at the ice edge region around Spitsbergen and Greenland. In January 2020 and February 2020 a stronger contribution to the ice growth originated from sea ice deformations driven by atmospheric wind forcing. The simulated ice divergence and ice shear have positive values in different regions of the BKS. There is negative ice divergence, which corresponds to convergence in the region of greatest sea ice growth, whereas the ice shear processes following wind stresses indicate here positive values.

From January 2020 until March 2020 low-pressure anomalies developed over the Eastern Arctic induced northerly winds from the cold Arctic Ocean to the BS and accelerated the southward drift of the MOSAiC ice floe in the Transpolar Drift. During March the low pressure anomalies were located north of the Laptev Sea, inducing westerly wind anomalies following Arctic cyclone tracks in the BKS and keeping the cold air in the Arctic. The simulated and satellite-derived sea ice velocity anomalies during January-March 2020 indicate a strong acceleration of the Transpolar Drift during the MOSAiC winter with intensified speeds up to 6 km/day. The drift of the Central MOSAiC Observatory was closely correlated with the zonal and meridional ERA5 10m wind components. Compared to previous years, winds tended to have anomalies toward the Fram Strait, in particular in January, February, and March 2020, in line with corresponding sea-level pressure patterns.

In the RASM hindcast, the Pan-Arctic thermodynamic ice volume tendencies increases due to ice growth from November until January. The unusual shift to a positive AO phase in the MOSAiC winter 2019/20 may indicate a strong portion of internal variability in sea ice formation in agreement with *Ding et al. (2019).* To quantify the underlying mechanisms for the Arctic ice volume differences and different temporal evolution we computed the thermodynamic sea ice volume tendencies with combined ice growth and ice melt terms for all 30 ensemble members in the forecast mode for the Pan-Arctic and the BS domains from November 2019-March 2020. In February 2020 and March 2020 the mean sea ice volume tendencies reaches 110 km$^3$/day in the Pan-Arctic and strong deviations occur due to internal variability in Arctic sea ice growth and dynamical ice deformations. The internally generated sea ice volume differences among the 30 ensemble members for the Arctic domain are in the range of 1000 km$^3$ and indicate strong internally generated variability due to Arctic feedbacks and remote impacts from the mid-latitudes. Some ensemble members develop positive ice volume differences and others negative differences relative to the ensemble mean. The great positive sea ice thickness anomaly in the Barents Sea during winter 2020 was connected to an enhanced ice growth following the colder temperature anomalies in this area, and a result of greater sea ice convergence and ice shears.

The differences between the hindcast and the four selected forecast simulations 2, 4, 8 and 9 are largest in November 2019 and December 2019 and can reach ~20 km$^3$/day, or ~600 km$^3$/month. For selected members of the model ensemble the dynamical contributions due to atmospheric advection and thermo-dynamic growth processes have been computed, which show the largest differences in the BS region. The integrated winter ice volume tendencies following dynamic and thermodynamic drivers are largest in the BS and linked to surface turbulent heat and momentum fluxes and oceanic convergence.

During the negative AO winter 2010, sea ice growth was enhanced in the Beaufort- and the Siberian seas and in a belt from the north coast of Greenland Sea to the Canadian Arctic with sea ice differences greater than 1 m compared to the positive AO winter 2020. Inverse temperature anomaly pattern occur between the western and the eastern Arctic for positive and negative AO winters. In this positive AO winter the eastern Arctic is colder and the western Arctic in the Canadian Basin warmer with an inversed pattern for the negative AO winter, connecting the AO phase to the AD pattern, discussed by *Watanabe et al. (2006).* The surface heat fluxes for JFM 2010 and JFM 2020 points to much stronger heat fluxes for the positive AO winter phase 2020 in the North Atlantic Ocean south of Greenland, whereas in the BS region and on the Pacific side the patterns look similar. This result supports the idea of *Sato et al. (2014),* that sea ice changes in the BS are under the control of atmospheric circulation over the Norwegian Seas and an enhanced southerly atmospheric advection connected to the northward shift of the Gulf stream, which influences the temperature and sea ice extent in the BS via a northward shifted North Atlantic storm track, which needs more in-depth investigation.

**Acknowledgement**

This work was carried out as part of the Multidisciplinary drifting Observatory for the Study of Arctic Climate (MOSAiC)
funded by the German Ministry for Education and Research (BMBF) under grant N-2014-H-060_Dethloff.

The production of the merged CryoSat-SMOS sea ice thickness data v2.02 was funded by the ESA project SMOS & CryoSat-2 Sea Ice Data Product Processing and Dissemination Service, and data from November 2010 to April 2020 were obtained from AWI (ftp://ftp.awi.de/sea_ice/product/cryosat2_smos/v202/). The ESA CryoSat-2/SMOS data set was produced and disseminated by the Alfred Wegener Institute, Helmholtz Centre for Polar and Marine Research.

The ERA5 data were provided by the European Centre for Medium-Range-Weather Forecasts (ECMWF).

Sea ice thickness data used in this manuscript were produced as part of the international Multidisciplinary drifting Observatory for the Study of the Arctic Climate (MOSAiC) with the tag MOSAiC20192020 and Project ID AWI_PS122.

WM, YL, JC and JCK acknowledge partial support of the RASM contribution from the following programs: the U.S. Department of Energy (DOE) Regional and Global Model Analysis (RGMA), the Office of Naval Research (ONR) Arctic and
Global Prediction (AGP) and National Science Foundation (NSF) Arctic System Science (ARCSS), and RO from the Ministry of Science and Higher Education in Poland. The U.S. Department of Defense (DOD) High Performance Computer Modernization Program (HPCMP) provided computer resources for RASM simulations analyzed here.

KD, DH and AR  acknowledge the funding by the Deutsche Forschungsgemeinschaft (DFG, German Research Foundation) – project number 268020496 – TRR 172, within the Transregional Collaborative Research Center "ArctiC Amplification:
Climate Relevant Atmospheric and SurfaCe Processes, and Feedback Mechanisms (AC)[3]".

TK, DH and AR acknowledge the project "Quantifying Rapid Climate Change in the Arctic: regional feedbacks and large-scale impacts (QUARCCS)" funded by the German and Russian Ministries of Research and Education.

**Competing interests**

The authors declare no competing interests.

**Code/Data availability**

Codes or data are available form the authors by request.

**Author contributions**

K. D. and W. M. conceived the study and wrote the paper. S. H., Y. L., H. F. G., T. K., C. H., D. H., and R. R. undertook the data analysis and developed the methods.  V. B., J. J. C., J. C. K., R. O., M. R., A. R., J. S., and A. S. contributed to the interpretation of results. All authors commented on the manuscript.

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

|  |  | Nov 2019 | Dec 2019 | Jan 2020 | Feb 2020 | Mar 2020 |
|---|---|---|---|---|---|---|
| CS2-SMOS | Mean (m) | 1,14 | 1,17 | 1,24 | 1,41 | 1,61 |
|  | S.D. | 0,71 | 0,66 | 0,65 | 0,71 | 0,77 |
| RASM | Mean (m) | 0,96 | 1,06 | 1,23 | 1,45 | 1,63 |
|  | S.D. | 0,59 | 0,56 | 0,57 | 0,58 | 0,61 |
| Corr. Coef |  | 0,74 | 0,74 | 0,75 | 0,76 | 0,74 |
| Bias |  | -0,18 | -0,1 | -0,01 | 0,04 | 0,01 |
| RMSD |  | 0,52 | 0,46 | 0,43 | 0,47 | 0,51 |


**Table 1.** Comparison of mean sea ice thickness (Mean), standard deviation (S.D.), correlations, bias and root mean square error (RMSD) between CryoSat2/SMOS satellite data and RASM simulations for November 2019 until March 2020.


|  |  | Nov 2019 | Dec 2019 | Jan 2020 | Feb 2020 | Mar 2020 |
|---|---|---|---|---|---|---|
| CS2-SMOS | Mean (m) |  |  |  |  |  |

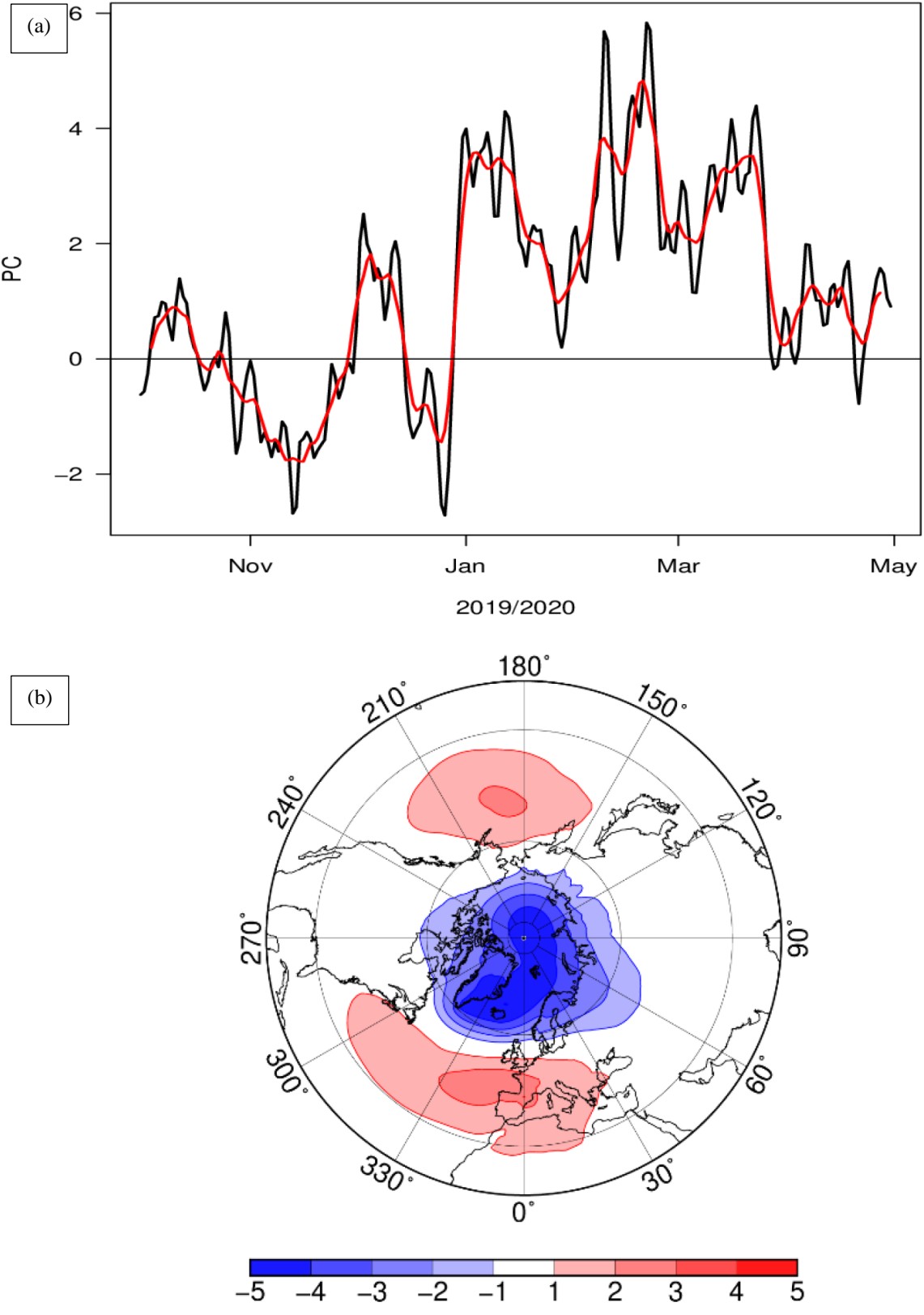

**Figure 1**: (a) Time series of daily values of the AO index from October 2019 to April 2020 (black line) with 7-day running mean (red line) and (b) the spatial AO pattern from 1979-2000 based on ERA5.

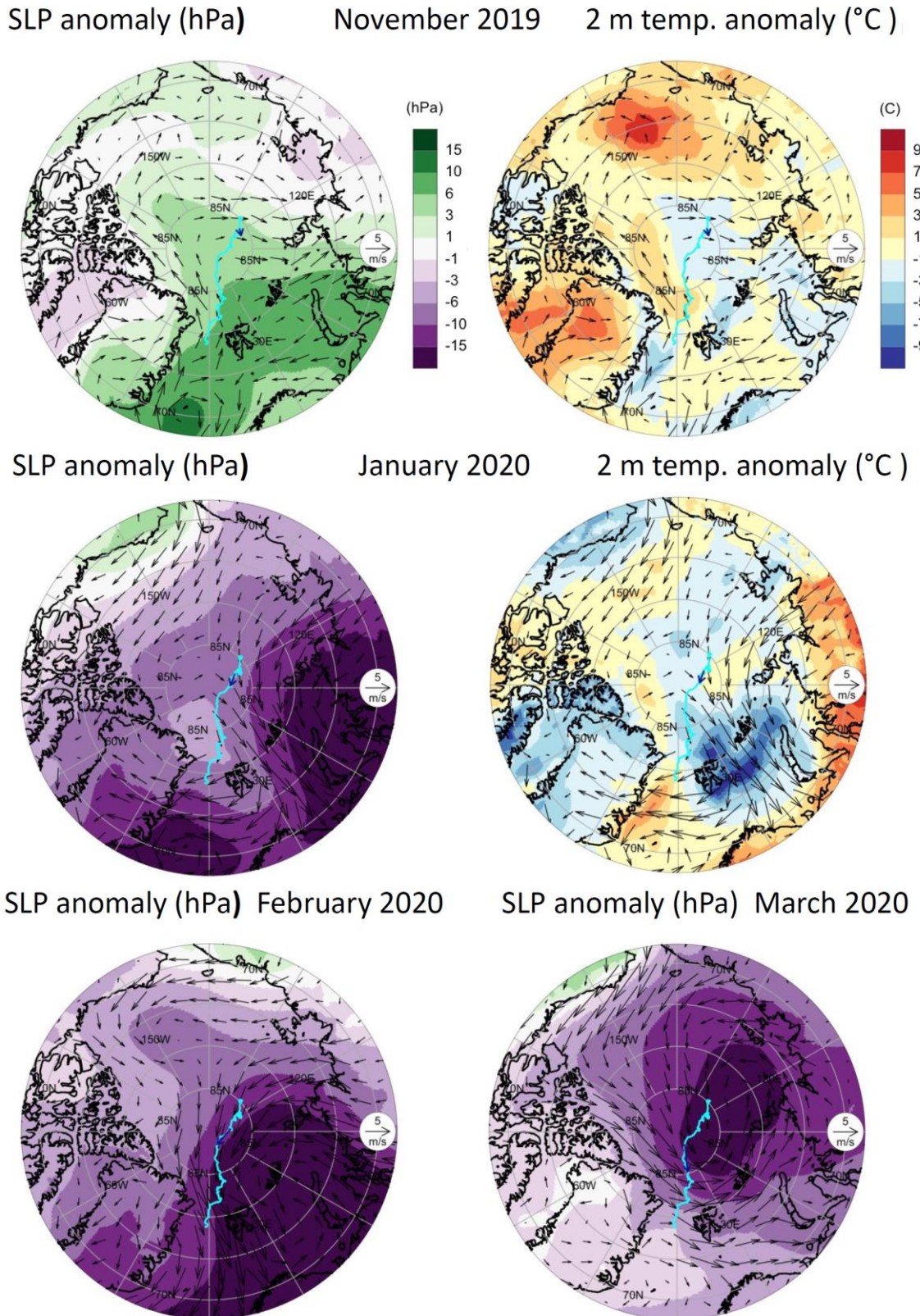

**Figure 2**. Sea level pressure anomaly (hPa) (left) and 2 m temperature anomaly (K) (right) compared to the climate mean 2010-2019 for November 2019 (top row) and January 2020 (middle row) based on ERA5. Sea level pressure anomalies (hPa) for February 2020 (left) and March (right) 2020 (bottom row). Arrows display the direction and strength of 10 m atmospheric winds. The cyan lines indicate the MOSAiC ice floe track from October 2019 until August 2020. Small blue arrows indicate the MOSAiC location and drift of the respective month.

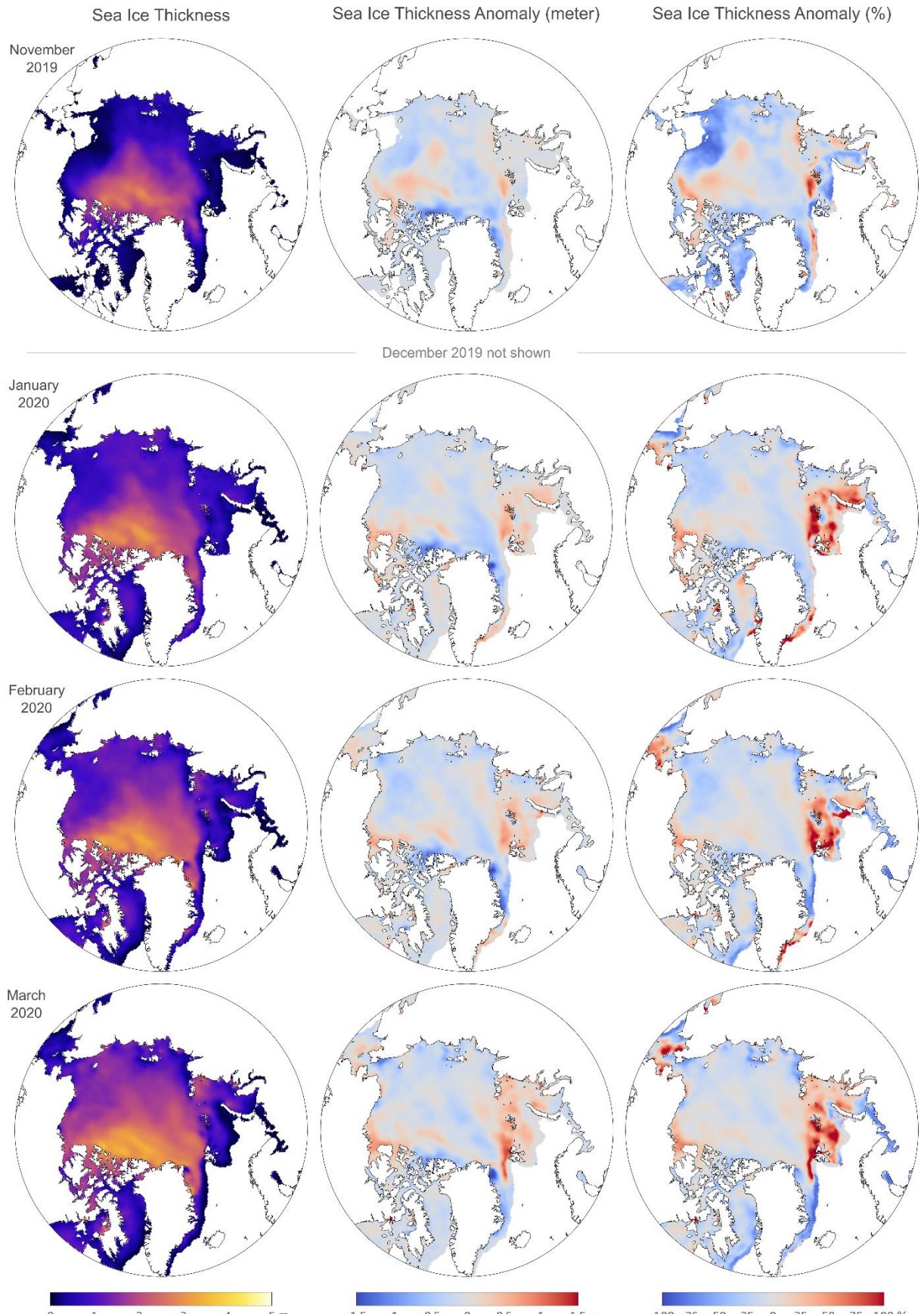

**Figure 3.** Sea ice thickness (left) and anomaly in meter (middle) and anomaly in percent (right) for November 2019 (top) through March 2020 (bottom). (December 2020 is not shown) based on CryoSat-2/SMOS satellite data analysis compared to the mean condition in the entire data record (2010-2019).

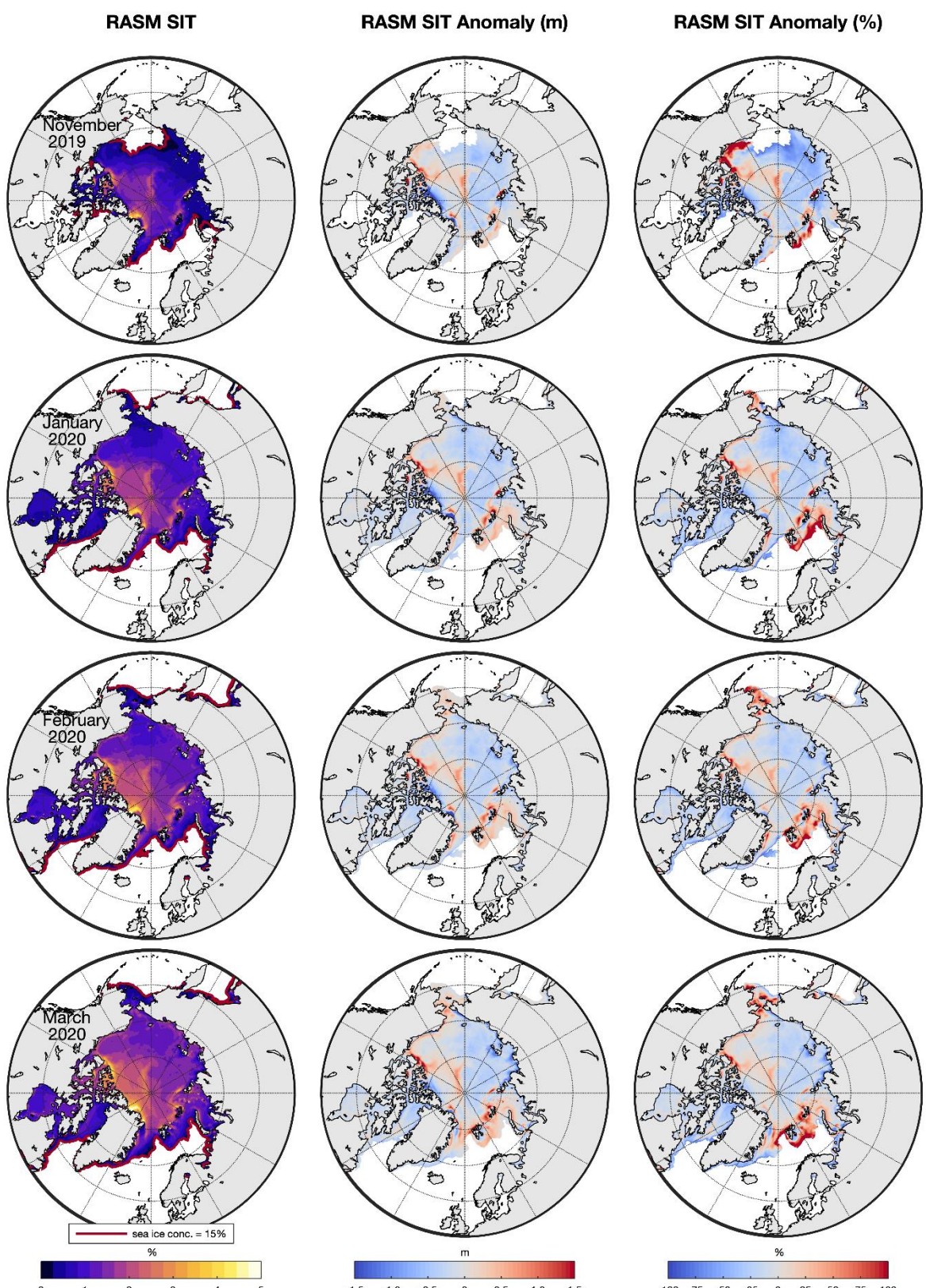

**Figure 4.** Sea ice thickness (m; black countour line for 15% sea ice concentration; left) and anomalies: in meter (middle), in percent (right) for November 2019 (top), January 2020 (2nd row), February 2020 (3rd row), and March 2020 (bottom) from the RASM hindcast simulation compared to the climate mean 2010-2019.

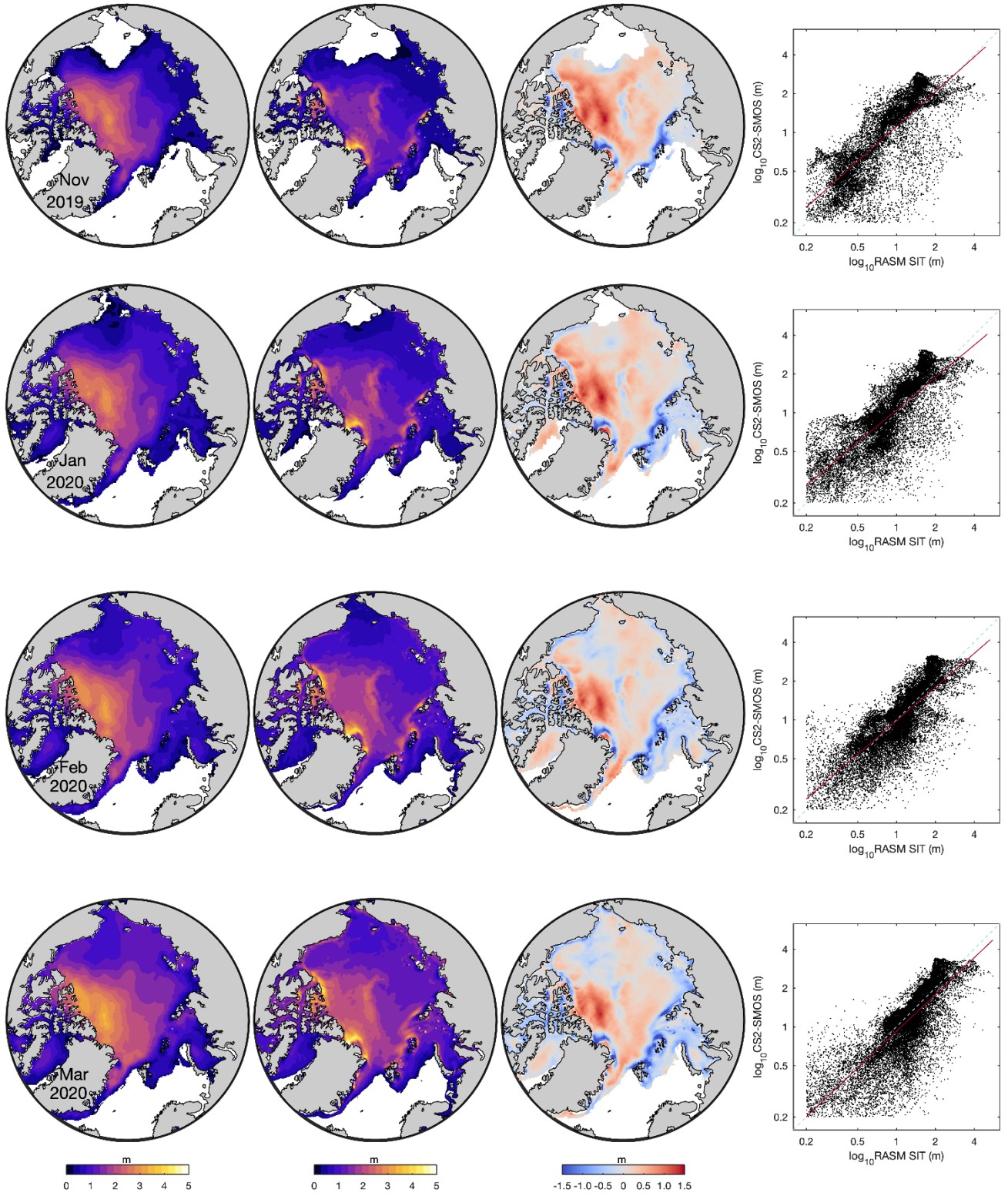

**Figure 5**. Sea ice thickness (m) for November 2019 (top), January ( 2nd row) and February 2020 (3rd row) and March 2020 (bottom) based on CryoSat2/SMOS (left column), RASM simulations (2nd column) and the differences "CryoSat2/SMOS minus RASM" (3rd column). The right column shows the scatter and correlation plots between CryoSat2/SMOS data and RASM results.

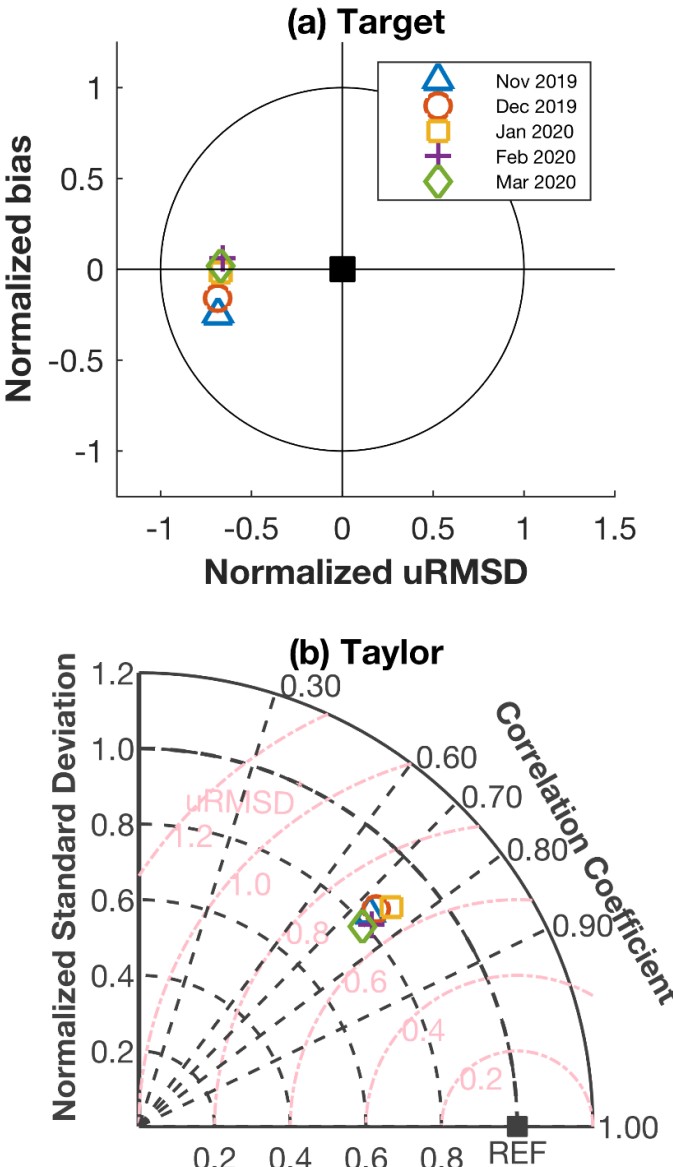

**Figure 6.** Target diagram (a) of normalised bias and normalized unbiased root-mean-square difference (uRMSD) and Tayloer diagram (b) of normalised standard deviation and correlation between the RASM sea ice thickness simulations and CryoSat2/SMOS data from November 2019 to March 2020. The square marker indicates the reference (REF) value, i.e., perfect model.

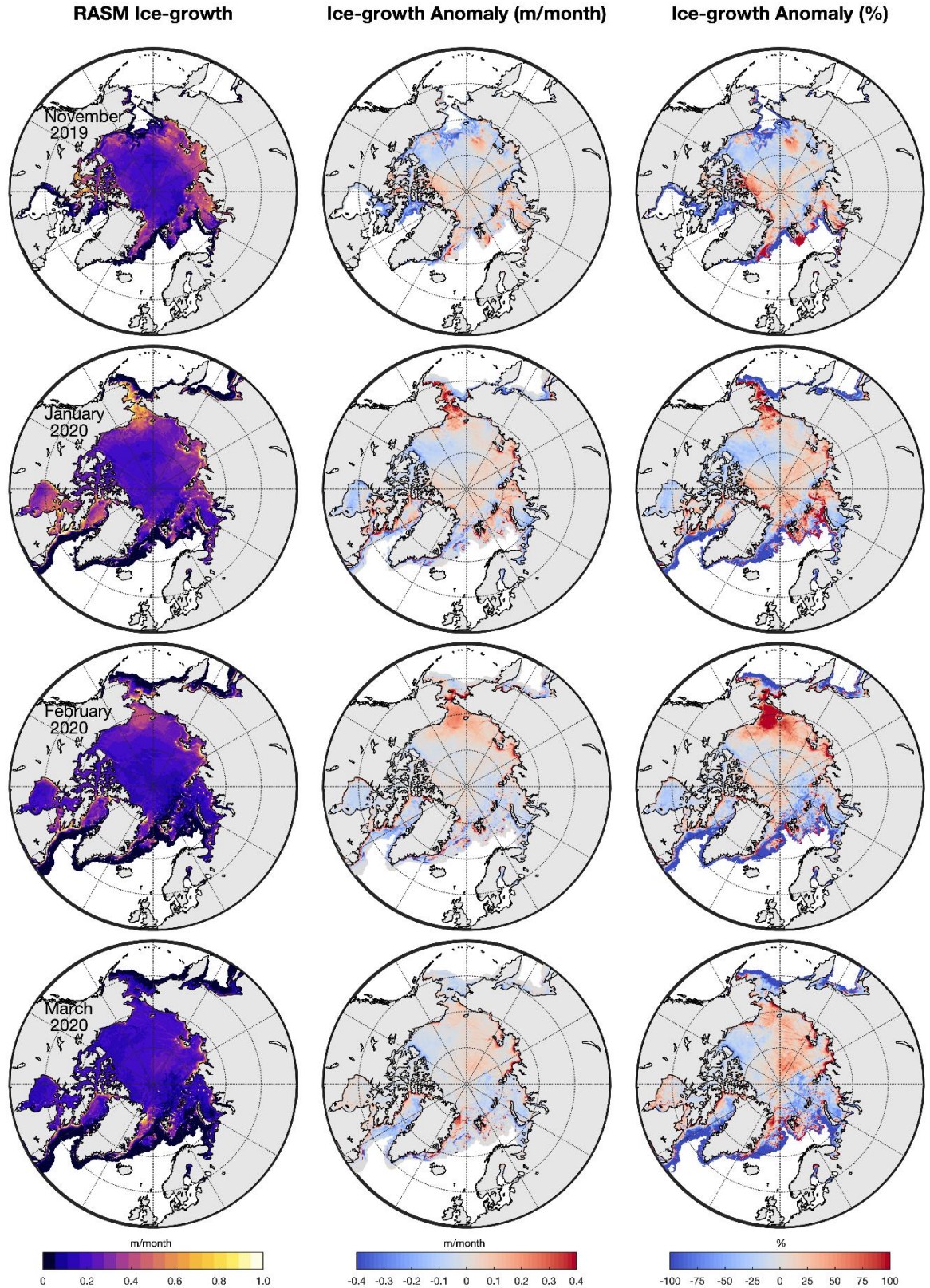

**Figure 7.** The integrated sea ice growth (m/month; left) and sea ice growth anomalies in m/month (middle) and in percent (right) for November 2019 (top), January 2020 (2nd row), February 2020 (3rd row), and March 2020 (bottom) from the RASM hindcast simulation compared to the climate mean 2010-2019.

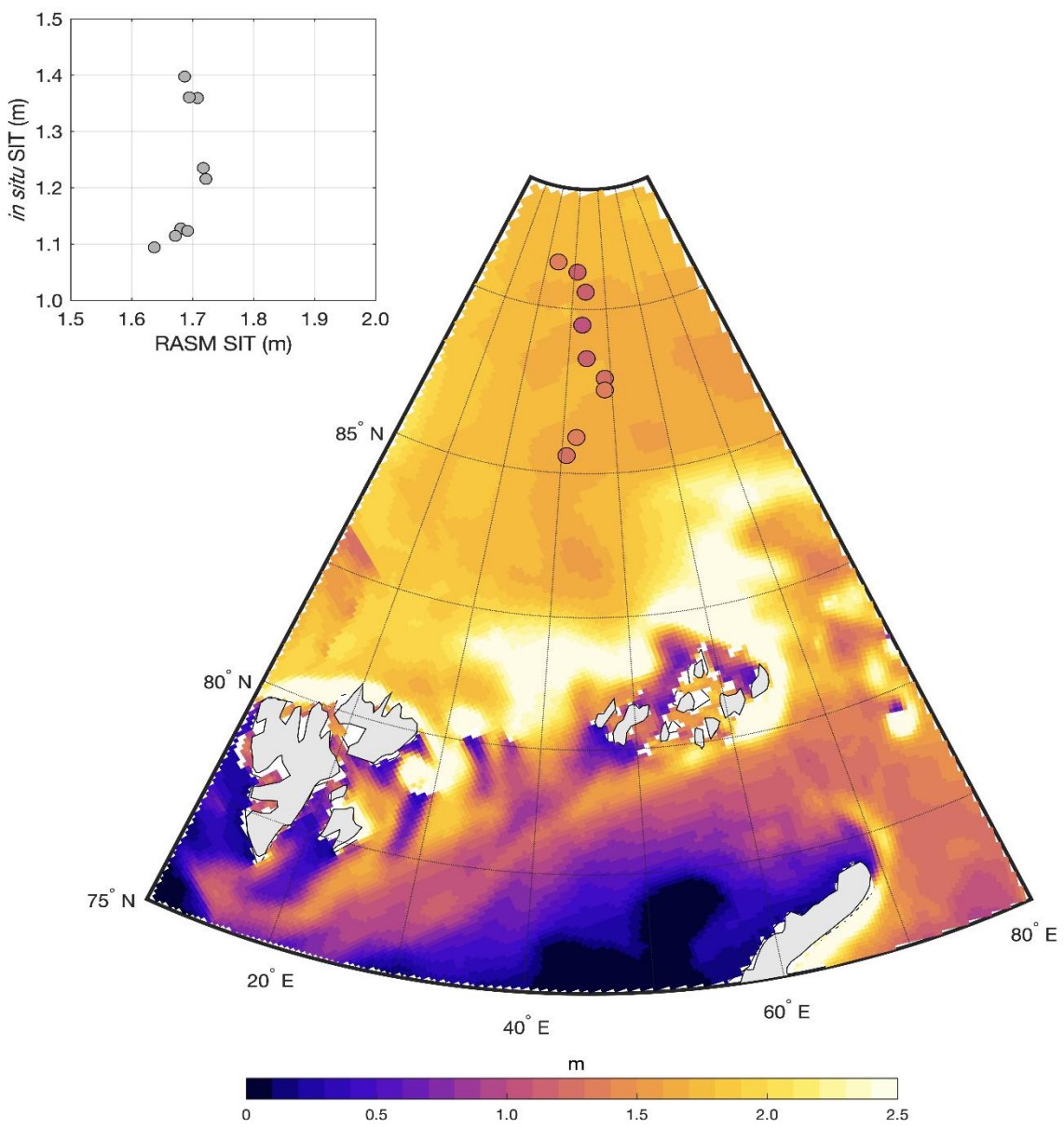


**Figure 8.** Mean sea ice thickness (m; shading) from the RASM hindcast simulation during March 6-14 and daily EM ice thickness measurements (m; circles) on "RV Kapitan Dranitzyn" from 6-14 March 2020. The small black picture (left) compares the in situ sea ice thickness measurements (m) with the corresponding RASM simulations at all points indicated by circles in the coloured plot (right).


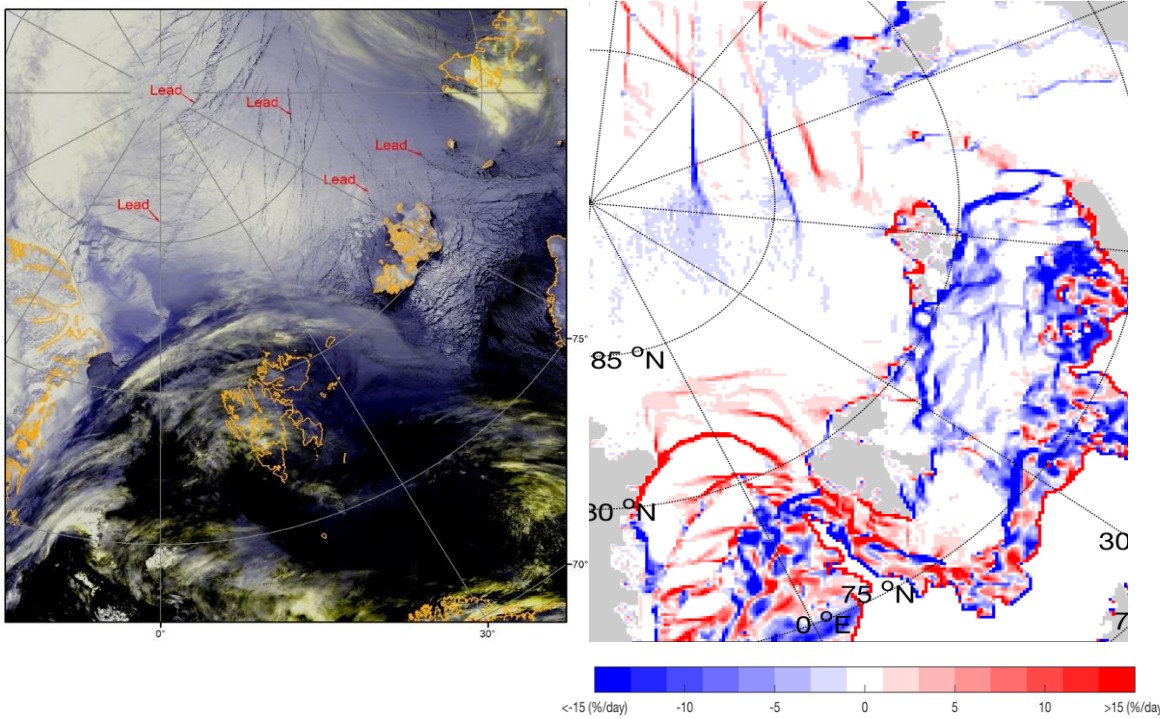

 **Figure 9.** Sea ice lead structures in data from infrared channel 5 of the NOAA-20 satellite with the highest possible resolution (375 m) on 5th March 2020 with identification of leads. These data were obtained using the VIIRS instrument (Visible / Infrared Imager Radiometer Suite), installed on board the NOAA-20 satellite (left). Daily mean sea ice divergence (%/day) on 5th March 2020 in the RASM simulation (right). No data manipulation was done except the daily averaging.





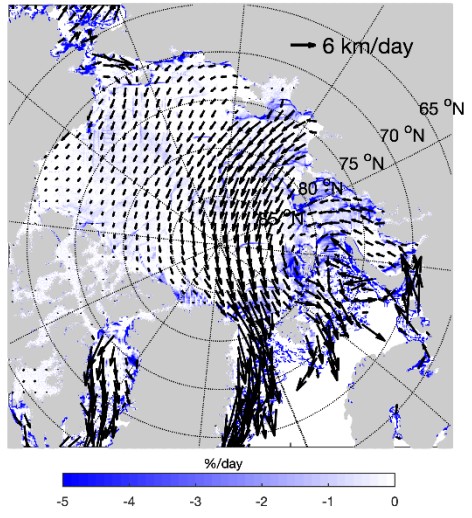

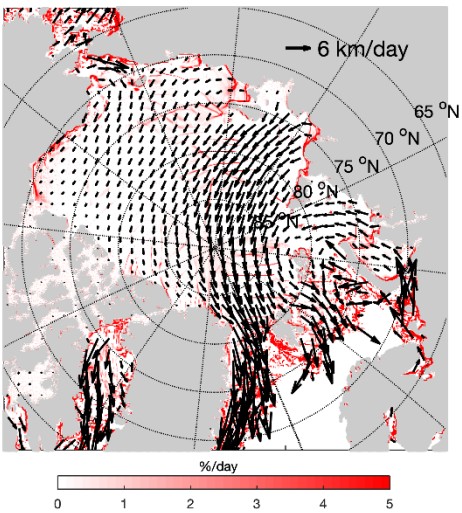

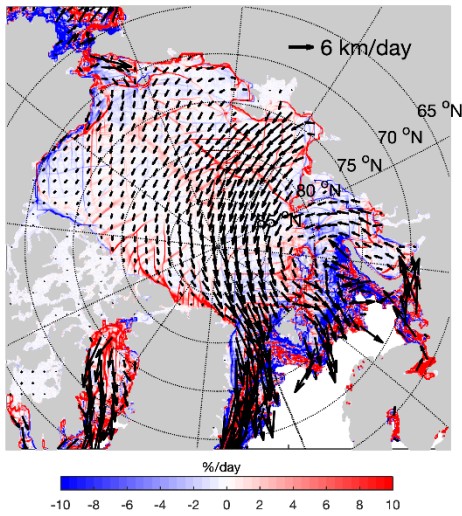

**Figure 10.** Sea ice anomalies calculated from the RASM hindcast simulation for January-March (JFM) 2020 compared to the RASM climate mean JFM 2010-2019 for negative divergence (top; blue shading represents less divergence) and positive divergence (middle; red shading represents more divergence). The bottom graph displays ice shear anomalies (%/day). The mean velocity vectors for the same period of JFM 2020 are overlaid in each plot.


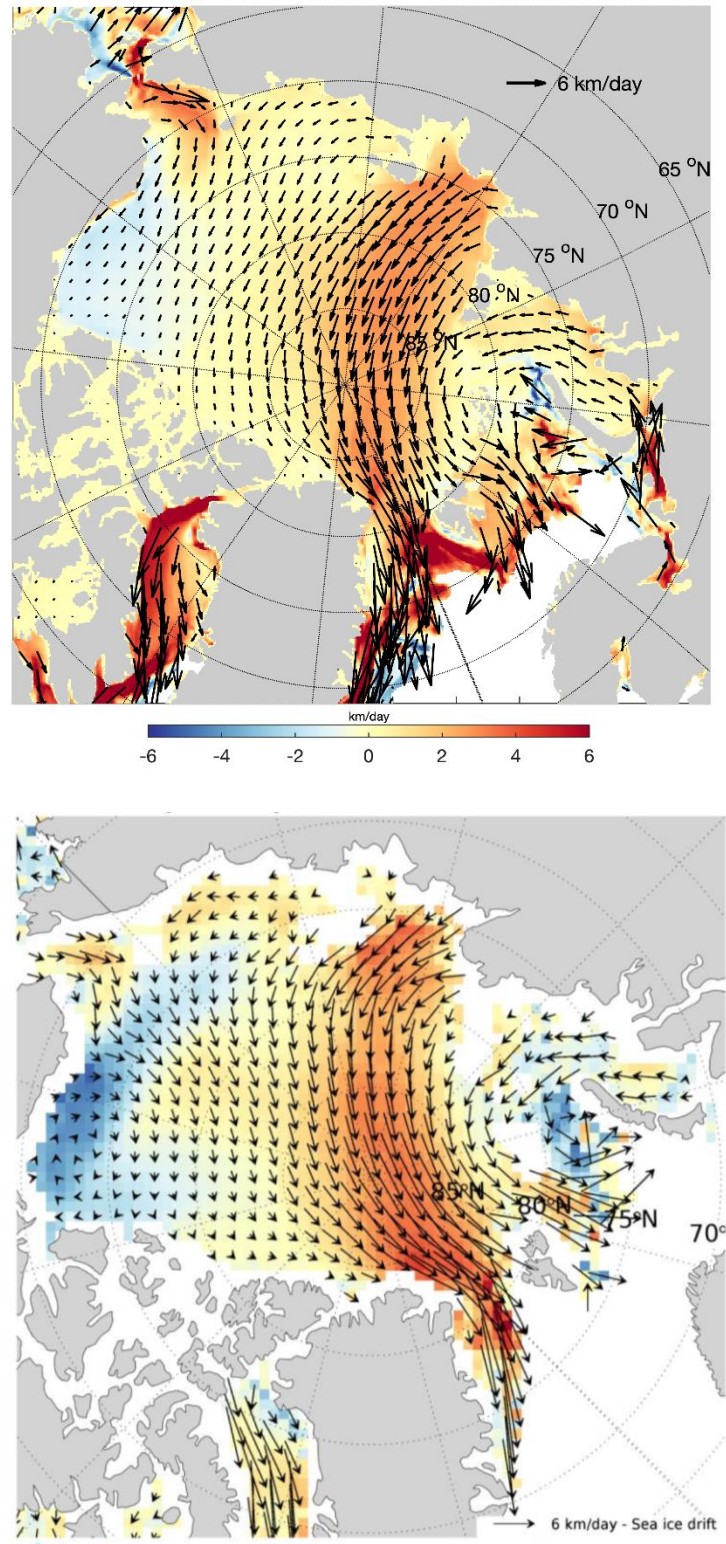

**Figure 11.** RASM simulations of sea ice velocity anomaly (top) and satellite derived sea ice velocity anomaly (km/day) 1010    (bottom) for January 2020-March 2020 compared to the climate mean 2010-2019.

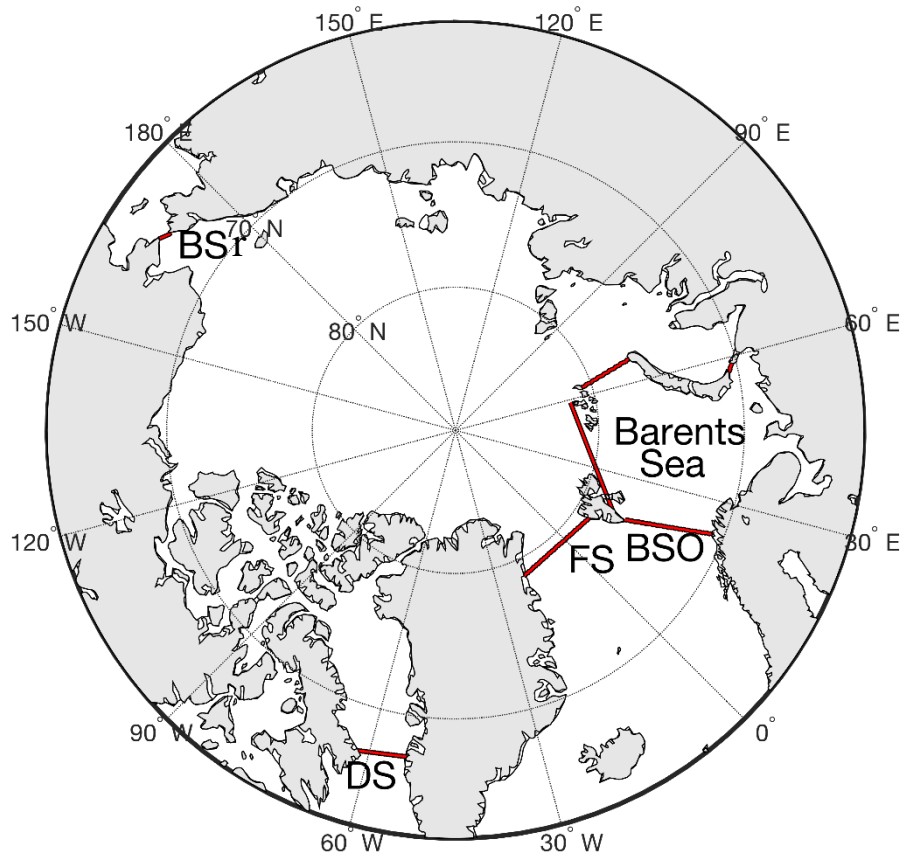


**Figure 12.** Pan-Arctic and Barents Sea domains used for the computation of the sea ice volume tendencies in Fig. 13.

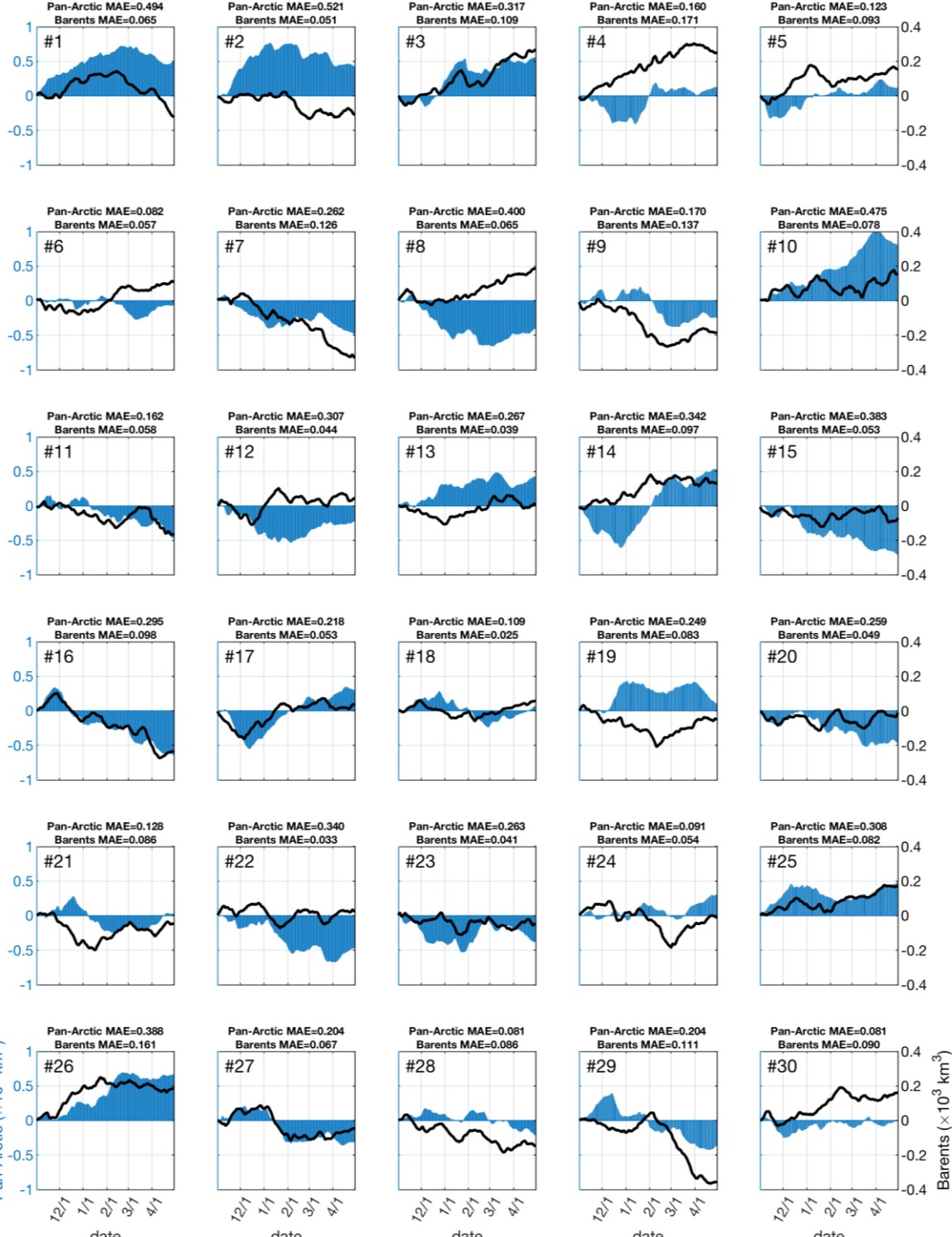

**Figure 13.** Temporal evolution of mean absolute difference of sea ice volume simulations ($10^3$ km$^3$) relative to the ensemble mean for 30 ensemble members of RASM integrations from 1th November 2010 until 30 th April 2020 in forecast mode for the Pan-Arctic domain (Blue) and the BS domain (black lines). Note the different Pan-Arctic (left) and BS (right) y-scales used in the panels.

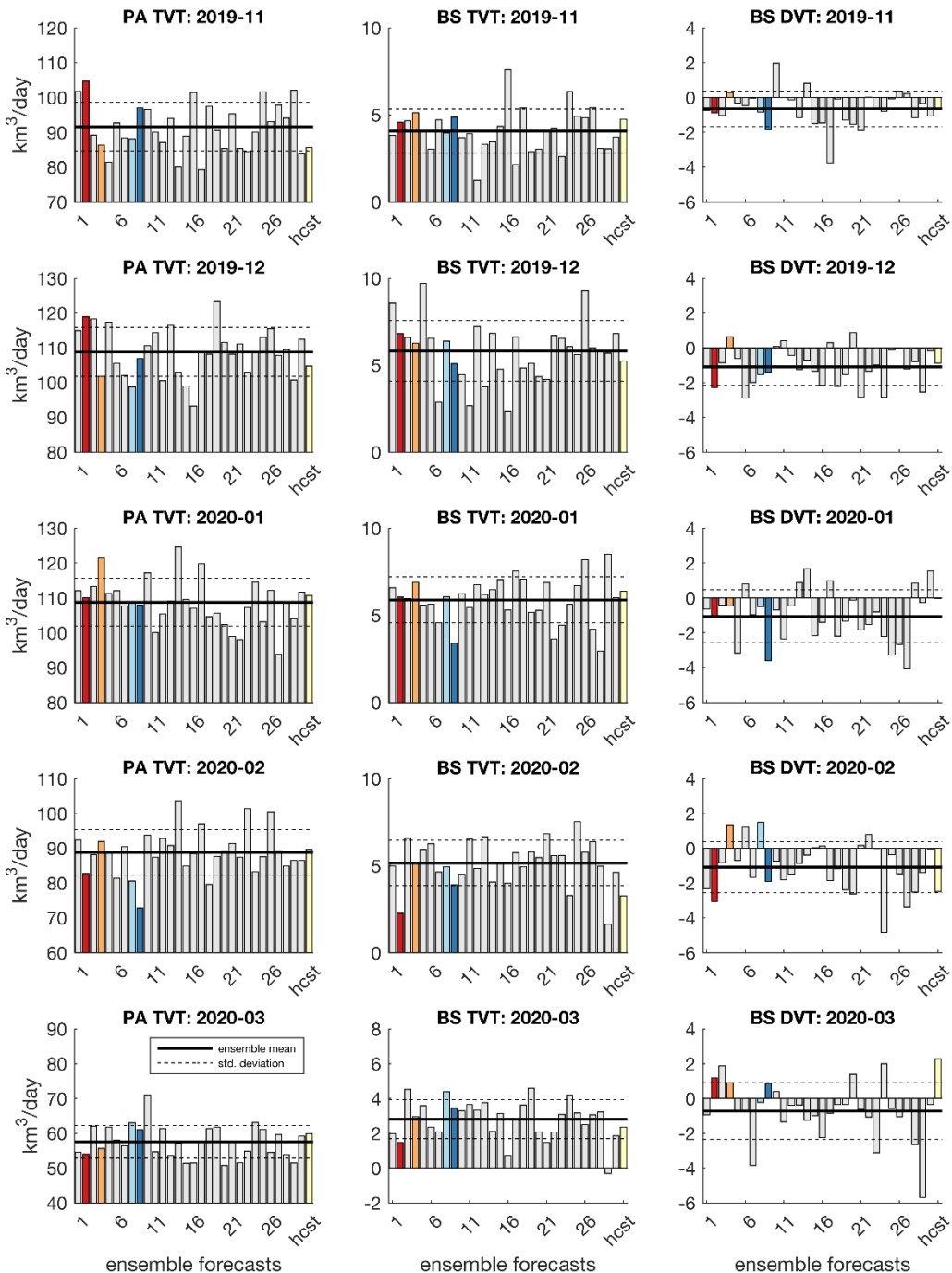

**Figure 14.** Thermodynamic sea ice volume tendencies (km³/day) of all 30 ensemble members and the RASM hindcast
simulation (yellow) for the Pan-Arctic (left) and the Barents Sea (middle) with the combined ice growth and ice melt terms
and the dynamical sea ice volume tendencies (km³/day) for the Barents Sea (right). Four individual ensemble forecast members
are selected and marked as member 2 (red), member 4 (orange), member 8 (light blue) and member 9 (dark blue).

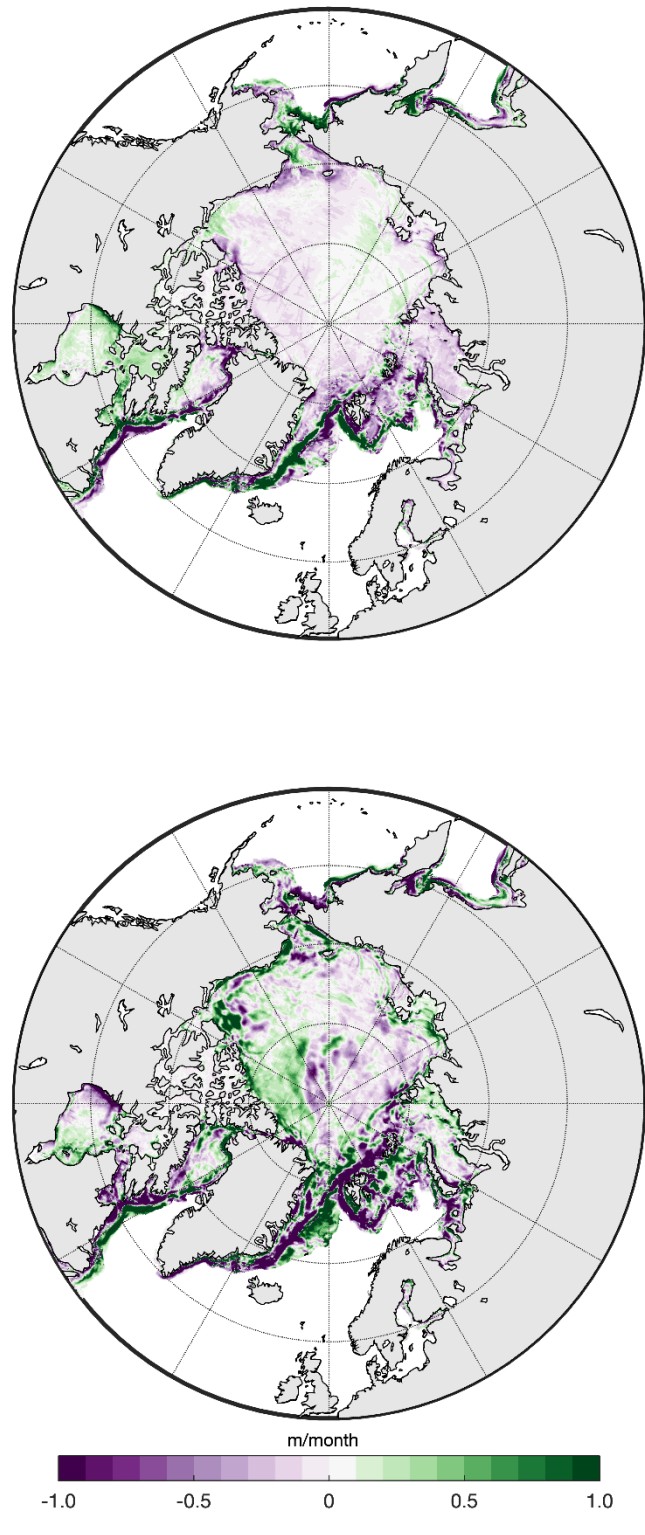

.**Figure 15.** Differences of accumulated thermo-dynamical  (m/winter) (top) and dynamical sea ice volume tendencies
1040    (m/winter) (bottom) for JFM 2020 between the RASM ensemble members 2 and 8.

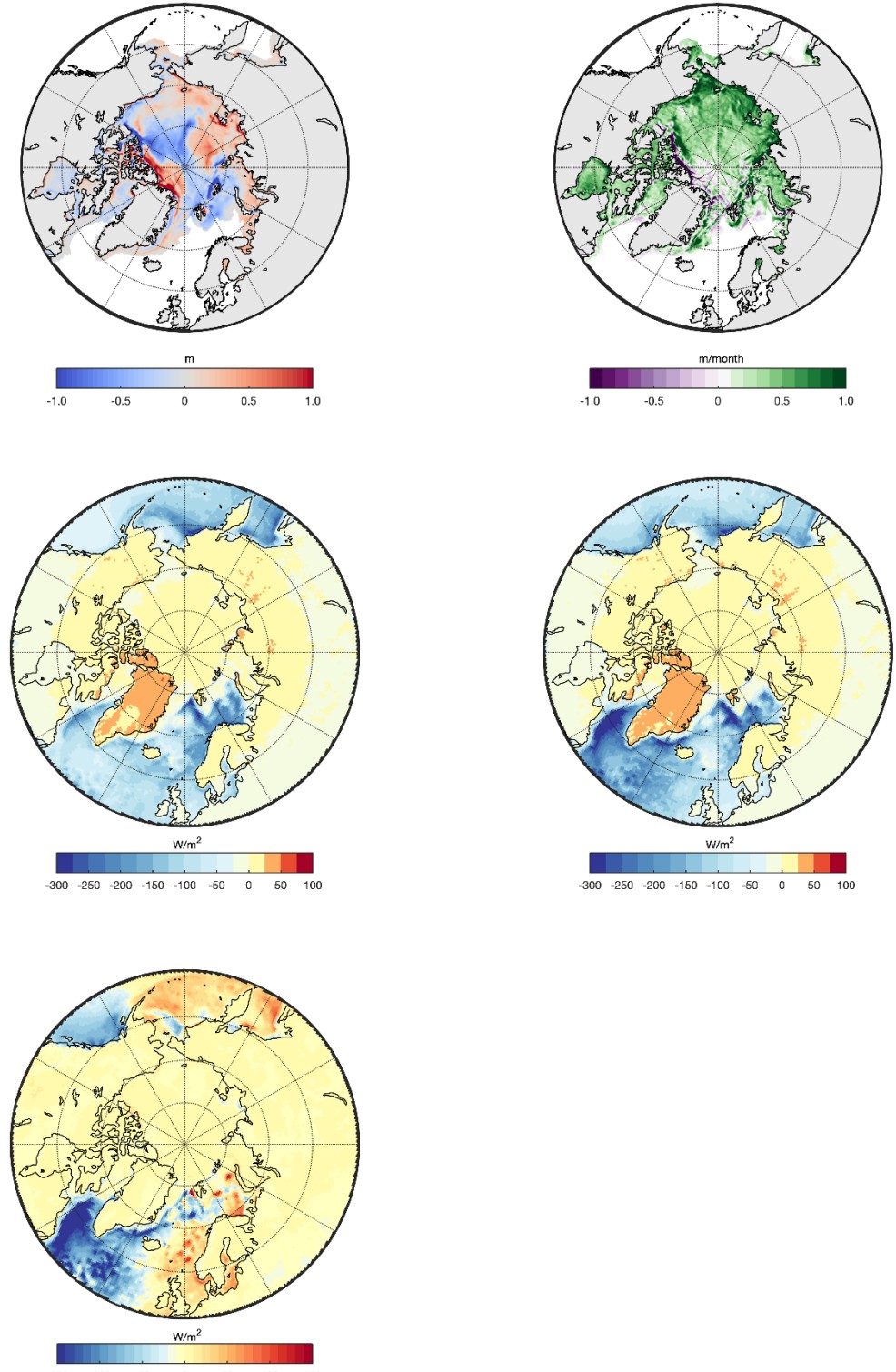

**Figure 16**. The differences of SIT (upper row, left) and total volume tendencies (upper row, right) "JFM 2010 minus JFM 2020" and the mean combined sensible and latent heat fluxes (W/m$^2$) for JFM 2010 (middle row, left) and for JFM 2020 (middle row, right) and the differences of the surface heat fluxes "JFM2020 minus JFM 2010" (lower row) from the RASM hindcast simulation. Note that the flux convention means, that negative fluxes are from the ocean into the atmosphere.