# Peer review of "Arctic sea ice anomalies during the MOSAiC winter 2019/20"

_The Cryosphere, 2020_

## Author Response (AR1)

**Answer to editor**

We thank the editor for her carefull reading of the manuscript and constructive remarks. In the manuscript we identified all changes in response to the editor suggestions in **red bold**.

**General comments**

We tried to improve the discussion of our results and the summary and conclusion section and incorporated all suggestions to streamline the text.

**LN: 223**
**The RASM ensemble forecast simulations were carried out from 1$^{st}$ November 2019 through 30$^{th}$ April 2020. Each ensemble member was initialized with the same sea ice and ocean conditions and on the same date, but then it was forced by a different, 24-hr apart, NCEP forecast data set which was initialized at 00.00 between 1st and 31st October 2019. The 30-member RASM ensemble was forced with different lateral boundary conditions from 9-months forecast of the NCEP climate forecast system, applying a linear nudging of temperature, zonal- and meridional wind above 500 hPa.**

**LN: 624**
**The internally generated sea ice volume differences among the 30 ensemble members for the Arctic domain are in the range of 1000 km$^3$ and indicate strong internally generated variability due to Arctic feedbacks and remote impacts from the mid-latitudes. Some ensemble members develop positive ice volume differences and others negative differences relative to the ensemble mean. The great positive sea ice thickness anomaly in the Barents Sea during winter 2020 was connected to an enhanced ice growth following the colder temperature anomalies in this area, and a result of greater sea ice convergence and ice shears.**

**Minor:**

**We accepted all suggestions and corrected all sentences.**

**The SIMS measurements were independent of the ASSIST protocol, but in general agreement with the visual observations. However, the ASSIST observations are prone to large errors because visual ice thickness observations were near to impossible due to the darkness of the polar night and due to the 100% ice coverage which prevented broken floes from tipping over.**

**We would like to keep the preview of upcoming chapters.**

**December 2019 was not shown simply to save space.**

**The underestimation of the ship based EM measurements and the correction of biases due to extensive ramming is under way and subject of another study by Haas et al. (in prep.) about the midwinter voyages of "RV Kapitan Dranitzyn" which we hope to submit by the end of the year.**

We replotted Fig. 10 and show convergence, divergence and shear anomaly plots.

We replotted Fig. 11 with the same isoline intervals.

We introduced the requested citations.

We removed the redundant subcaptions in Fig. 5, 10, 11, 15 and 16.

We would like to keep the subcaptions in Fig. 7.

We would like to keep also Fig. 12 in accordance with one reviewer.

Leads have been visually identified in the VIIRS.

Fig. 9b shows the RASM daily mean sea ice divergence simulated on 3/5/2020, No data manipulation was done except the daily averaging, which is performed within the CICE code for specified temporal output.

The three panels of Fig. 10 show sea ice anomalies calculated from the RASM hindcast simulation for January-March (JFM) 2020 compared to the RASM climate mean JFM 2010-2019 for negative divergence (top; blue shading represents less divergence) and positive divergence (middle; red shading represents more divergence). The bottom graph displays ice shear anomalies (%/day). The mean velocity vectors for the same period of JFM 2020 are overlaid in each plot.

**Answers to reviewer 1**

We thank the reviewer for his/her constructive remarks. In the manuscript we identified all changes in response to the reviewer in **black bold**.

**General comments**

Figures 4, 7, and 16 were replotted and enlarged as suggested.

Figures 2, 3, and 5 were replotted to be similar to Fig. 12 as suggested.

Figs. 10 and 11 are vector plots and a polar-stereographic projection doesn't work well due to the convergence of meridians toward the pole. Fig. 10 was replotted.

We explained all abbreviations about the Target and Taylor diagramms in Fig. 6 by improved figure caption: **Figure 6. Target diagram (a) of normalised bias and normalized unbiased root-mean-square difference (uRMSD) and Taylor diagram (b) of normalised standard deviation and correlation between the RASM sea ice thickness simulations and CryoSat2/SMOS data from November 2019 to March 2020. The square marker indicates the reference (REF) value, i.e., perfect model.**

We introduced the following sentences for description of the Target and Taylor diagramms: **RASM skill is assessed using the Target diagram (Joliff et al., 2009) to visualise root-mean-square difference (RMSD; distance from a center), unbiased RMSD (uRMSD; x-axis), and bias (y-axis) for monthly SIT on a single plot. They are normalised by the standard deviation of CryoSat2/SMOS SIT. The Taylor diagram (Taylor, 2001) provides an additional set of statistics in uRMSD by displaying the correlation and the ratio of the standard deviation between RASM and CryoSat2 SIT**

We added the following citations:

**Jolliff, J. K., et al. ,2009, Summary diagrams for coupled hydrodynamic ecosystem model skill assessment, J. Mar. Syst., 76, 64–82.**
**Taylor, K. E., 2001, Summarizing multiple aspects of model performance in a single diagram, J. Geophys. Res., 106, 7183–7192.**

Summary and conclusion sections were partly revised.

**Specific comments:**

LN 165-167: We described the concept in the text**. The concept is described by *Ricker (2020)* in the CryoSat2-SMOS merged product description document. An optimal interpolation scheme (OI) has been used, that allows the merging of datasets from diverse sources on a predefined analysis grid. The data are weighted differently based on known uncertainties of the individual products and an estimated correlation length scale. OI minimizes the total error of observations with respect to a background field and provides ideal weighting for the observations at each grid cell. The background field consists of a weighted average of CryoSat-2 and SMOS data two weeks before and after the rolling observation period with a length of 7 days. The CryoSat2-SMOS product is then defined as the sea-ice thickness analysis fields of the 7 day observation period with the center date as the reference time of each file. Melting does not allow to**

retrieve sea-ice thickness estimates from CryoSat-2 and SMOS during summer between May and September. Therefore, the merged product is limited to the period from mid-October to mid-April only due to the background field requirement.

LN 208: Title of section 3.1 was changed to: **3.1 Analysis of atmospheric and sea ice conditions in ERA5 and satellite data**

LN 216: In Fig. 1a we explain the AO time series and in Fig. 1b the spatial AO pattern.

We explain with the following text why Fig. 1b is based on ERA-5 1979-2000.

**Figure 1a presents daily values of the AO index in mean sea level pressure (SLP) based on ERA-5 from October 2019 until May 2020 with 7-day running mean (red line) and Fig. 1b the spatial AO pattern north of 20 °N. The AO pattern was defined as the leading mode of Empirical Orthogonal Function analysis of monthly mean SLP during the 1979-2000 period over the domain 20°-90°N. This domain and reference period was used for the calculation of the spatial AO patterns to ensure a comparability with the widely used AO index provided by the NOAA Climate Prediction center (CPC, https://www.cpc.ncep.noaa.gov/products/precip/CWlink/daily_ao_index/ao.shtml), which is based on the AO pattern calculated for the mentioned reference period and NCEP/NCAR reanalysis data set. The daily AO indices (Fig. 1a) have been obtained by projecting ERA5 daily SLP data from 1979 to May 2020 onto the AO pattern shown in Fig. 1b. For comparison the loading pattern of the AO for the different ERA-5 reference period 2010-2019 was computed (not shown), and the corresponding AO index for the MOSAiC period, obtained by projecting the daily SLP anomalies onto this loading pattern. The time series of daily values of the AO index from October 2019 to April 2020 obtained by projecting the daily SLP anomalies onto the loading pattern from 2010-2019 agree entirely with Fig 1a.**

For the information of the reviewer (not presented in the manuscript), we display in Fig. R1a the AO pattern for the period 2010-2019 and in Fig. R1b the AO time series of by projecting onto the loading pattern shown in Fig. R1a.

[Figure]

Fig. R1a.Time series of daily values of the AO index from October 2019 to April 2020 (black line) with 7-day running mean (red line), obtained by projecting the daily SLP anomalies onto the loading pattern shown in Fig. R1b.

[Figure]

Figure R1b. Loading pattern of AO based on monthly mean SLP for the period 2010-2019 (ER5 data).

LN 219: Fig. 2 was revised

LN 255-257: Figs. 3 and 4 have been replotted using the same color bars. The mentioned sentence was changed to: **The largest positive thickness anomalies between 1.0 and 1.5 m occur in the BS, along the north-eastern Canadian coast and in the central Arctic Ocean.**

LN 265-267: The sentence was changed to:

**The differences in SIT (Fig. 5) may be partly connected to the impact of surface roughness on the radar freeboards and the retrieval algorithms as discussed by Landy et al. (2020). Specifically, the merged CryoSat-2/SMOS SIT data is dominated by the CryoSat-2 radar altimeter contribution in areas with multi-year sea ice. Radar freeboard is the term in sea ice radar altimetry that describes the height of the ice surface above local sea level perceived by a radar altimeter. It differs from sea ice freeboard, the actual height of the ice surface, by a correction that requires prior knowledge of snow depth and density. The purpose of this correction is to remove the impact of the slower wave propagation speed of the radar pulse within the snow lawyer on the radar range and thus ice surface elevation. The corrected radar freeboard is then converted to sea ice thickness using information of the densities of sea ice, ocean water, and snow, and estimating the depth of snow accumulated on the ice surface based on climatological values (Hendricks et al. 2020). In Landy et al. (2020) it is demonstrated that sea ice surface roughness may cause a systemic radar freeboard uncertainty which represents one of the principal sources of pan-Arctic SIT uncertainty. In the CryoSat-2 retrieval algorithm of the CryoSat-2/SMOS SIT data set this systemic bias might contribute to the higher CryoSat-2/SMOS thicknesses in the central Arctic and specifically north of the Canadian Archipelago with respect to RASM in Fig 5. But this assertion does not consider other systemic uncertainties present in the CryoSat-2 retrieval such as the underestimation of sea ice density for multi-year ice in recent years (Jutila et al.**

**2021), which might compensate the radar freeboard bias to an unknown extent. In the comparison to other SIT data sets, CryoSat-2/SMOS also yields thicker ice in the central Arctic compared to ICESat-2 estimates and though these difference are within the range of the SIT uncertainty resulting from different retrievals, an indication of SIT overestimation by CryoSat-2/SMOS remains.**

LN 274: **The domain averaged bias is the difference between RASM and satellite data in the region shown in Fig. 12, including all the ice regions with boundaries at DS (Davis Street), FS (Fram Strait), BSO (Barents Sea Opening) and BS (Bering Strait).**

LN 277: The sentence was changed to: **The integrated sea ice growth anomalies of RASM compared to the mean 2010-2019, are displayed in Figure 7.**

LN 285-286: The sentence was changed to: **The EM ice thickness measurements on board of the Russian icebreaker „RV Kapitan Dranitzyn" as part of the MOSAiC resupply between**…

LN 288-290: The sentence was changed to: **The main bias of the EM measurements is connected to the difficulty of instrument calibration on the ramming icebreaker. The frequent ramming operations of the ship with little progress over the undisturbed heavy ice makes processing and filtering of the ship-based measurements challenging**.

LN297-301: We replotted the satelite picture in Fig. 9 to show the leads north of Spitsbergen more clearly and added red arrows with the word ‚Lead'.

LN 313-329: We revised the 1st part of this paragraph as follows:

**Figure 10 displays the anomalies of ice convergence (top), of divergence (middle) and in the bottom panel the ice shear anomaly (all in percent/day) simulated by RASM for the JFM 2020 mean compared to the JFM 2010-2019 mean. In all three plots the transpolar drift in km/day is indicated by thick black arrows. Longer black arrows in the Davis Strait, the east coast of Greenland and the BS indicate individual grid cells with a very different drift. Blue colours in the top part of Fig. 10 indicate regions with reduced convergence and red colours in the middle part of Fig. 10 indicate those with enhanced divergence. These grid cells are likely reflecting the free-drift of thinner sea ice in marginal ice zones, where the impact of atmospheric wind forcing on the ice drift is much less limited compared to the drift within pack ice.**

LN 333: OSI-SAF is explained in the text as follows: **The Ocean Sea Ice Satellite Application Facilities (OSA-SAF) deliver satellite derived scatterometer winds, sea surface temperatures and sea ice surface temperatures, radiative fluxes, sea ice concentration, edges, types and sea ice drift.**

Fig. 11 was revised. **The black arrows over the redish shading in the Eastern Arctic indicate the transpolar drift. Longer black arrows in the Davis Strait, the east coast of Greenland and the BS indicate the free-drift of grid cells within marginal ice zones.**

LN: 350: Changed to Bering Strait.

LN 363 and 369: Abbreviations were removed and physical processes are described.

LN: 401-403: Fig. 15 was replotted and enlarged. The mentioned differences at the ice edge region are now clearly visible.

We added the following sentence: **Although the greatest differences occur at the ice edge regions, remarkable changes in the central Arctic, north-west of Greenland, are visible and sea ice volume in this region is to a large extent determined by dynamical processes.**

LN: 416-419:

We added the following sentence:

**Comparison of Fig. S12 with Fig. S6 indicates an inverse temperature anomaly pattern between the western and the eastern Arctic for positive and negative AO winters. Under positive AO conditions in winter (Fig. S6 exemplarily for January 2020)) negative temperature anomalies occur over the eastern Arctic and positive anomalies occur over the Canadian Basin in the western Arctic. During the negative AO winter conditions in January 2010 (Fig. S12) the eastern Arctic reveals weak positive temperature anomalies and the western Arctic negative temperature anomalies.**

We thank the reviewer for his/her constructive remarks. In the manuscript we identified all changes in response to the reviewer in **blue bold.**

**General comments:**

**Point 1:**

We followed the suggestion of the reviewer and described anomalies of the atmospheric MOSAiC observed parameters along the drift trajectory (10 m wind, 2m temperature, sea level pressure) compared to the climatology 2010-2019 in section 3.1.1 and **Figure S4** and meridional sea ice velocities together with the 10 m zonal and meridional wind components in **Figure S5.**

**The text was changed accordingly:**

[revised manuscript text omitted]

We added additional citations in the references.

**Specially comments:**

1) These results have been described in a recently submitted paper Krumpen et al. (2021) and cited here.

2) We found it difficult and prefer to keep our wording with thermodynamic processes and combined effects of thermodynamical and dynamical changes.
3) LN 310: Changed to "**the SIT distribution in a region**".
4) LN 330: Changed to "**the climate mean of 2010-2019**".
5) LN 389: Changed to "**Compared to the short daily atmospheric time scales the longer time scales of ocean and sea ice processes provide memory effects for seasonal sea ice forecasts…**"
6) LN 430: Changed to "**losing heat to the atmosphere**"

---

## Author Response (AR2)

**Answer to the editor**

We thank the editor again for her carefull reading of the manuscript. All ms changes in response to the editor suggestions are in red.

**General comments**

We rewrote the section part from line 407-416.

We revised the "Summary and conclusion" section completely.

We corrected the mentioned lines.

Figure captions were increased.

We enlarged Fig. S2.

**Minor**

We moved the data source into the Acknowledgement and added a sentence to thank the reviewers and the editor.

We corrected all lines according to the suggestions of the editor.

**Detailed comments on editors questions:**

LN 405: RASM uses an energy-conserving ocean model with a rotated sphere mesh with the equator extending across the North Pole, resulting in a ca 9 km resolution in the Arctic Ocean. Therefore no divergence pattern problem at the pole occurs.

LN 407-416 were rewritten see above.

LN 424-438 were rewritten.

LN 466 was improved.

LN 480: Volume tendencies have been explained.

LN 496 was better explained.

LN 481-526 were rewritten. The presented Fig. 14 was specifically developed to capture a metric of thermodynamical and dynamical changes of the ensemble members.

LN 555-571: We improved the text. The suggested changes in energy budget effects on TD and Dyn are beyond the scope of this paper, since they would require additional ensemble simulations for that winter.

Conclusions 573-637 were rewritten completely, see above.